# Active regulation of the epidermal growth factor receptor by the membrane bilayer

Shwetha Srinivasan, Xingcheng Lin, Xuyan Chen, Raju Regmi[†], Bin Zhang*, Gabriela S Schlau-Cohen*

Department of Chemistry, Massachusetts Institute of Technology, Cambridge, United States

## eLife Assessment

The authors describe an interesting approach to studying the dynamics and function of membrane proteins in different lipid environments. The **fundamental** findings have theoretical and practical implications beyond the study of EGFR to all membrane signalling proteins. The evidence supporting the conclusions is **compelling**, based on the use of a nanodisk system to study membrane proteins in vitro, combined with state-of-the-art single-molecule FRET. The work will be of broad interest to cell biologists and biochemists.

*For correspondence:
binz@mit.edu (BZ);
gssc@mit.edu (GSS-C)

Present address: [†]Institut Curie, CNRS, Laboratoire Physico Chimie Curie, Paris, France

**Abstract** Cell surface receptors transmit information across the plasma membrane to connect the extracellular environment to intracellular function. While the structures and interactions of the receptors have been long established as mediators of signaling, increasing evidence suggests that the membrane itself plays an active role in both suppressing and enhancing signaling. Identifying and investigating this contribution has been challenging owing to the complex composition of the plasma membrane. We used cell-free expression to incorporate the epidermal growth factor receptor (EGFR) into nanodiscs with defined membrane compositions and characterized ligand-induced transmembrane conformational response and interactions with signaling partners using single-molecule and ensemble fluorescence assays. We observed that both the transmembrane conformational response and interactions with signaling partners are strongly lipid dependent, consistent with previous observations of electrostatic interactions between the anionic lipids and conserved basic residues near the membrane adjacent domain. Strikingly, the active conformation of EGFR and high levels of ATP binding were maintained regardless of ligand binding with high anionic lipid content typical of cancer cells, where EGFR signaling is enhanced. In contrast, the conformational response was suppressed in the presence of cholesterol, providing a mechanism for its known inhibitory effect on EGFR signaling. Our findings introduce a model of EGFR signaling in which the lipid environment can override ligand control, providing a biophysical basis for both robust EGFR activity in healthy cells and aberrant activity under pathological conditions. The membrane-adjacent protein sequence, likely responsible for the lipid dependence, is conserved among receptor tyrosine kinases, suggesting that active regulation by the plasma membrane may be a general feature of this important class of proteins.

## Introduction

The epidermal growth factor receptor (EGFR), the canonical receptor tyrosine kinase, maintains basic cellular processes in mammals (*Lemmon and Schlessinger, 2010*; *Yarden and Sliwkowski, 2001*).

EGFR phosphorylates adaptor proteins that trigger an array of signaling cascades responsible for cell proliferation and differentiation or, upon aberrant activity, leads to disorders such as cancer and fibrosis (*Yarden and Pines, 2012*; *Chen et al., 2016*). Ligand binding to the extracellular region of the receptor initiates a signaling response that propagates across the plasma membrane (*Lemmon et al., 2014*). The composition of the membrane changes from healthy to disease states in a manner known to influence EGFR activity (*Desai and Miller, 2018*; *Casaletto and McClatchey, 2012*; *Du and Lovly, 2018*; *Kim et al., 2021*). A basic understanding of the ligand-induced structural reorganization of the receptor has been established (*Ogiso et al., 2002*; *Garrett et al., 2002*; *Lemmon, 2009*; *Zhang et al., 2006*; *Zhang et al., 2007*; *Arkhipov et al., 2013*; *Shan et al., 2012*; *Shan et al., 2013*; *Srinivasan et al., 2022*), yet the contribution of the plasma membrane in regulating this reorganization remains underinvestigated (*Bessman and Lemmon, 2012*).

The chemical composition of the human plasma membrane is complex. The membrane composition is well maintained in healthy cells, yet becomes dysregulated in a diseased state, where EGFR signaling is often aberrant (*Spector and Yorek, 1985*; *Dias and Nylandsted, 2021*). In healthy cells, the plasma membrane is an asymmetric lipid bilayer containing 30% anionic lipids in the inner leaflet (*Lorent et al., 2020*). These anionic lipids modulate the surface charge of the membrane, affecting protein localization and clustering (*Noack and Jaillais, 2020*). Anionic lipids also act as signaling molecules and regulate several fundamental cellular processes (*Sunshine and Iruela-Arispe, 2017*; *Lee, 2004*; *Duncan et al., 2017*; *Qiu et al., 2018*; *Pond et al., 2020*). The anionic lipid content increases in cancer cells, where EGFR signaling is typically increased (*Ran et al., 2002*; *Vasquez-Montes et al., 2019*; *Stafford and Thorpe, 2011*; *Szlasa et al., 2020*). Similarly, dysregulation of lipid metabolism and distribution coupled with hyperactivation of EGFR are major hallmarks of neurodegenerative disorders, including Alzheimer's disease, Parkinson's disease, dementia, and sclerosis (*Wei et al., 2023*; *Lemkul and Bevan, 2011*). Cholesterol is the major sterol component of mammalian cell membranes, averaging 20–25% of the lipid bilayer (*Ikonen, 2008*). Cholesterol maintains the structural integrity and regulates the fluidity of the bilayer (*Chapman, 1975*; *Subczynski et al., 2017*). Increasingly, cholesterol has been implicated in the modulation of signal transduction and cellular trafficking (*Ohvo-Rekilä et al., 2002*; *Zhang et al., 2019*; *Koshy and Ziegler, 2015*). In the case of EGFR, cholesterol has been shown to suppress ligand-induced signaling (*Ringerike et al., 2002*). While a relationship between the plasma membrane and the signaling cascade has been established, how individual components, particularly anionic lipids and cholesterol, influence the protein-level transmembrane conformational response of EGFR remains unclear.

Several studies have investigated the role of anionic lipids and cholesterol for individual domains of EGFR. The plasma membrane was found to abrogate the catalytic activity of the intracellular kinase domain (*Endres et al., 2013*) and reduce the ligand binding affinity for an unliganded protomer in a dimer of the extracellular domain, leading to a negative cooperativity model (*Arkhipov et al., 2014*; *Alvarado et al., 2010*; *Macdonald and Pike, 2008*; *Liu et al., 2012*). Cholesterol has also been shown to impede phosphorylation for EGFR in vitro (*Ge et al., 2001*). In transmembrane-juxtamembrane constructs of EGFR, electrostatic interactions between the positively charged juxtamembrane region and the negatively charged anionic lipids have been identified (*Mineev et al., 2015*; *Doerner et al., 2015*; *Lelimousin et al., 2016*; *Bocharov et al., 2017*; *Jura et al., 2009*; *Bocharov et al., 2016*). While these studies have provided some understanding of the impact of the plasma membrane on individual domains, transmembrane conformational signaling requires propagation through these domains to cross the membrane. Piecewise studies inherently cannot probe how such propagation is regulated by membrane composition.

Here, we report the observation of a transmembrane conformational response of EGFR in membranes with different compositions using single-molecule Förster resonance energy transfer (smFRET), ensemble fluorescence assays, and molecular dynamics simulations. These measurements showed a robust ligand-induced conformational response in membranes that mimic healthy cells, consistent with previous observations (*Arkhipov et al., 2013*; *Srinivasan et al., 2022*). However, in membranes enriched with anionic lipids and/or cholesterol, we found a suppression of this response—revealing a previously unrecognized effect of these lipid components. These results indicate that membrane composition actively modulates EGFR function through both its charge and mechanics, playing a key role in robust signaling. Therapeutic development targeting EGFR may, therefore, require the context of the plasma membrane for optimal efficacy.

## Results and discussion

### EGFR in membrane nanodiscs

EGFR-containing discoidal membranes, termed 'nanodiscs', were produced by in vitro co-expression of EGFR and an apolipoprotein in the presence of lipid vesicles, leading to self-assembly of the structures shown in *Figure 1a*, *Figure 1—figure supplements 2 and 3*; *He et al., 2015*; *Quinn et al., 2019*. Nanodiscs were produced with different ratios of 1-palmitoyl-2-oleoyl-sn-glycero-3-phosphocholine (POPC), 1-palmitoyl-2-oleoyl-sn-glycero-3-phospho-L-serine (POPS), and cholesterol and dimyristoylphosphatidylcholine (DMPC; *Figure 1—figure supplements 4 and 5*). POPC is a zwitterionic phospholipid forming neutral membranes, whereas POPS carries a net negative charge and provides anionic character to the bilayer (*Her et al., 2016*). Both PC and PS lipids are common constituents of mammalian plasma membranes, with PC enriched in the outer leaflet and PS in the inner leaflet (*Lorent et al., 2020*). Nine lipid environments with different combinations of anionic lipid and/ or cholesterol content were compared: 0%, 15%, 30%, 60% POPS in POPC; 7.5%, 20% cholesterol in POPC and in 30% POPS in POPC; and DMPC, the fully saturated analog of the monounsaturated POPC. Zeta potential analysis was used to confirm the incorporation of POPS into the nanodiscs (*Figure 1b*, *Figure 1—figure supplement 6*; *Her et al., 2016*). Laurdan, a fluorescent membrane marker for membrane fluidity, was used to confirm the incorporation of cholesterol (*Figure 1c*, *Figure 1—figure supplement 7*; *Parasassi et al., 1994*).

A fluorescent donor-acceptor dye pair was incorporated into the nanodisc construct for fluorescence measurements. The donor dye (snap surface 488 or 594) was covalently attached to the snap tag at the EGFR C-terminus and the acceptor dye was introduced as a Cy5-labeled lipid at low concentration in the lipid bilayer or bound as Atto647N-labeled γ-ATP (*Figure 2—figure supplement 1*; *Quinn et al., 2019*; *Srinivasan et al., 2022*). For the Cy5-labeled lipid, the steady-state absorption and emission spectra, and lifetime of the acceptor dye were consistent between different nanodisc membrane compositions, indicating no lipid-dependent photophysics (*Figure 2—figure supplement 2*). The functionality of labeled EGFR in the nanodiscs was evaluated by Western blotting for phosphorylation of the tyrosine residues, which showed levels consistent with previously published assays on similar preparations (*Figure 1—figure supplement 2d*; *He et al., 2015*; *Quinn et al., 2019*; *Srinivasan et al., 2022*).

### Role of anionic lipids in EGFR kinase activity

The enzymatic activity of the catalytic site in the EGFR kinase domain can be regulated through its accessibility to ATP and other substrates (*Kovacs et al., 2015*). To investigate how the membrane composition impacts accessibility, we measured ATP binding levels for EGFR in membranes with different anionic lipid content. 1 μM of fluorescently-labeled ATP analogue, atto647N-γ ATP, which binds irreversibly to the active site, was added to samples of EGFR nanodiscs with 0%, 15%, 30%, or 60% anionic lipid content in the absence or presence of EGF. The fluorescence intensity from the bound ATP analogue and the fluorescence intensity from snap surface 488, which binds stoichiometrically to the snap tag at the EGFR C-terminus, were measured for each sample from a fluorescent gel image. The relative amount of ATP binding was quantified for each sample by normalizing to the EGFR content (*Figure 2a and b*).

How the level of ATP binding changed with anionic lipid content depended on the presence of EGF. In the absence of EGF, ATP binding was high in neutral bilayers (0% POPS) and in highly anionic (60% POPS) bilayers, but low at physiologically relevant POPS levels (15% and 30% POPS). This suggests that nucleotide binding is suppressed in the physiological regime, likely to ensure EGF can promote catalytic activity. In the presence of EGF, ATP binding overall increased with anionic lipid content with the highest levels observed in 60% POPS bilayers. In the neutral bilayer, ligand seemed to suppress ATP binding, indicating anionic lipids are required for the regulated activation of EGFR. Similar, and relatively high, levels of ATP binding were observed in the physiological regime, consistent with the model of EGF-promoted activity. The high ATP binding at 60% POPS —even without EGF —is consistent with the enhanced levels of catalytic activity characteristic of cancer cells, which show high anionic lipid content (*Ran et al., 2002*; *Vasquez-Montes et al., 2019*; *Stafford and Thorpe, 2011*; *Szlasa et al., 2020*). In previous work, experiments and molecular dynamics simulations identified electrostatic interactions between anionic lipids and the kinase domain of EGFR (*McLaughlin et al., 2005*; *Arkhipov et al., 2013*). These results highlight that membrane composition, through electrostatic

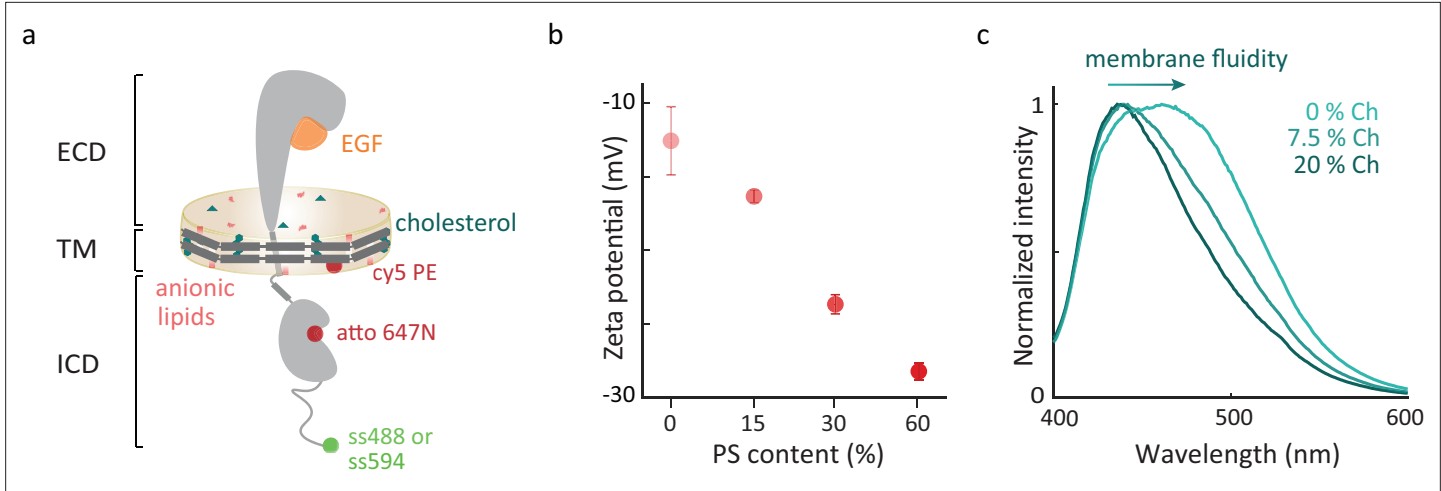

**Figure 1.** EGFR in different membrane environments. (**a**) Full-length EGFR (gray) embedded in a nanodisc. The nanodisc is a lipid bilayer (beige) belted by an amphiphilic apolipoprotein (dark gray). EGFR consists of a 618-amino-acid extracellular region that binds EGF (orange), a 27-amino-acid transmembrane-spanning domain, and an intracellular region, which is a 37-amino-acid juxtamembrane domain, a 273-amino-acid kinase domain, and a 231-amino-acid disordered C-terminal tail. Green and maroon spheres indicate the donor and acceptor dyes, respectively (*Figure 1—figure supplement 1*). (**b**) Mean of zeta potential distributions for EGFR in nanodiscs containing increasing amounts of anionic lipids (0%, 15%, 30%, and 60% POPS). Error bars are from three technical replicates. (**c**) Ensemble fluorescence emission spectra ($\lambda_{exc} = 385$ nm) of EGFR-embedded Laurdan containing nanodiscs with increasing cholesterol.

The online version of this article includes the following source data and figure supplement(s) for figure 1:

**Source data 1.** Raw data underlying *Figure 1b*.

**Source data 2.** Raw data underlying *Figure 1c*.

**Figure supplement 1.** Domains of EGFR.

**Figure supplement 2.** Production and characterization of full-length EGFR in nanodiscs.

**Figure supplement 2—source data 1.** PDF file containing original SDS-PAGE gels for *Figure 1—figure supplement 2b*, indicating the relevant bands and experimental conditions.

**Figure supplement 2—source data 2.** Original files for SDS-PAGE gels for *Figure 1—figure supplement 2b*.

**Figure supplement 2—source data 3.** PDF file containing original SDS-PAGE gels for *Figure 1—figure supplement 2c*, indicating the relevant bands and experimental conditions.

**Figure supplement 2—source data 4.** Original files for SDS-PAGE gels for *Figure 1—figure supplement 2c*.

**Figure supplement 2—source data 5.** PDF file containing original SDS-PAGE gels for *Figure 1—figure supplement 2d*, indicating the relevant bands and experimental conditions.

**Figure supplement 2—source data 6.** Original files for SDS-PAGE gels for *Figure 1—figure supplement 2d*.

**Figure supplement 3.** Optimization of ApoA1 and EGFR co-expression in cell-free reactions.

**Figure supplement 3—source data 1.** PDF file containing original SDS-PAGE gels for *Figure 1—figure supplement 3*, indicating the relevant bands and experimental conditions.

**Figure supplement 3—source data 2.** Original files for SDS-PAGE gels for *Figure 1—figure supplement 3*.

**Figure supplement 4.** Characterization of EGFR-containing nanodiscs in different anionic membrane environments.

**Figure supplement 4—source data 1.** Raw data underlying *Figure 1—figure supplement 4*.

**Figure supplement 5.** Characterization of EGFR-containing nanodiscs in membrane environments containing cholesterol.

**Figure supplement 5—source data 1.** Raw data underlying *Figure 1—figure supplement 5*.

**Figure supplement 6.** Characterization of anionic content in EGFR-embedded nanodiscs with zeta potential (*Her et al., 2016*).

**Figure supplement 6—source data 1.** Raw data underlying *Figure 1—figure supplement 6*.

**Figure supplement 7.** Characterization of cholesterol (Ch) content in EGFR-embedded nanodiscs with Laurdan.

**Figure supplement 7—source data 1.** Raw data underlying *Figure 1—figure supplement 7*.

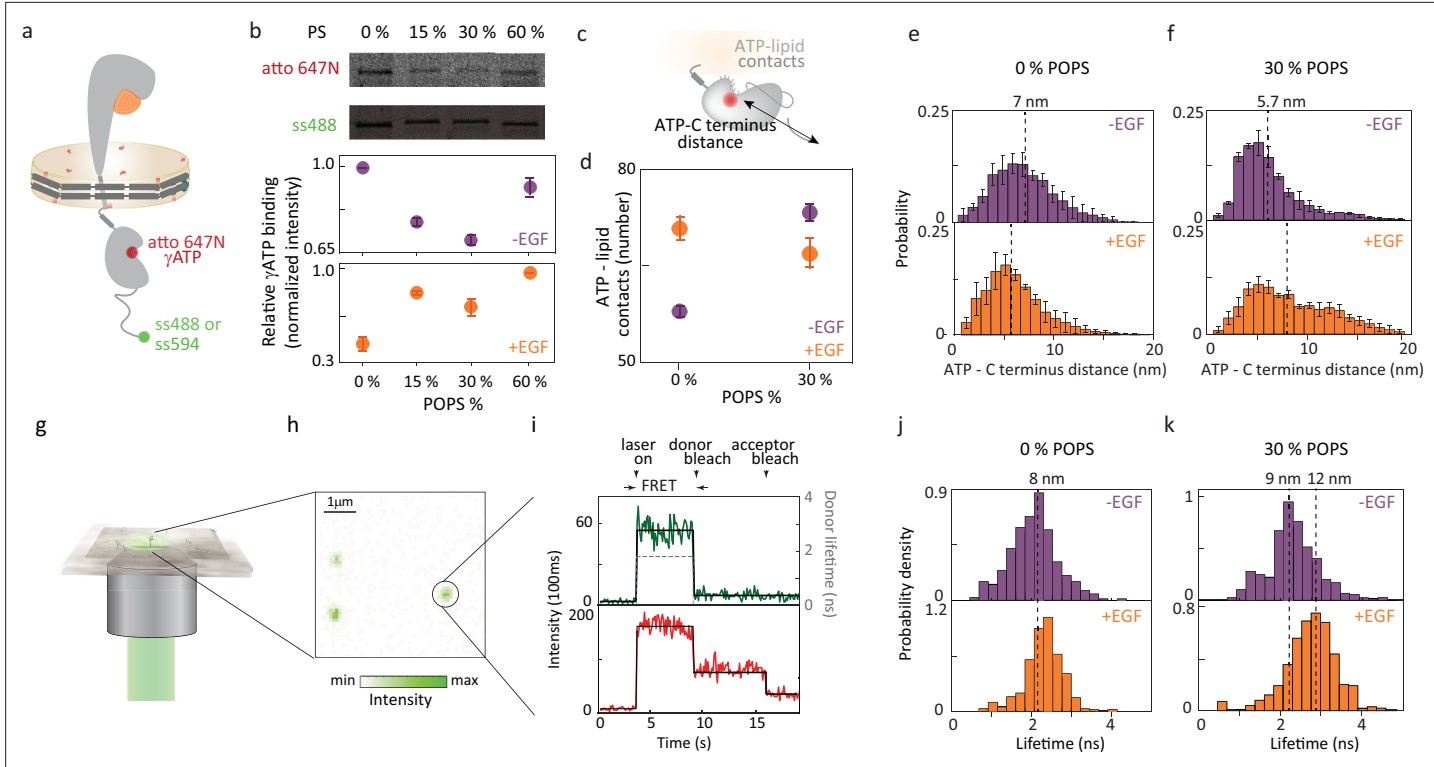

**Figure 2.** Membrane composition influences EGFR function through ATP binding. (**a**) Full-length EGFR in nanodiscs with atto647N γ ATP (red sphere) and snap surface 488 or snap surface 594 (green sphere). (**b**) Extent of ATP binding in different anionic environments quantified using the intensity of atto 647 N (top) band normalized by the amount of EGFR produced as extracted from the intensity of ss488 band (center) as a function of negatively charged POPS lipids (bottom) in the absence (purple) and presence (orange) of EGF. Error bars from three independent biological replicates. (**c**) EGFR intracellular domain indicating ATP binding site-C-terminus distance and ATP-lipid contacts measured from molecular dynamics simulations. (**d**) Accessibility of ATP binding site quantified through the contact number between the ATP binding site and lipids in the absence (purple) and presence (orange) of EGF. Error bars from three equal partitions of the simulations. Probability distributions of the distance between residue 721, the closest residue to the ATP binding site (**Honegger et al., 1987**), and EGFR C-terminus for (**e**) neutral (0% POPS) and (**f**) 30% anionic lipids (30% POPS) without EGF (top); with 1 μM EGF (bottom). Dotted lines indicate the medians on all histograms with corresponding distances on upper x-axis (**Table 1**). (**g**) Schematic of multiparametric single-molecule confocal microscope. (**h**) Fluorescence intensity for a representative image ($\lambda_{exc} = 550$ nm) where green spots are immobilized EGFR nanodiscs. (**i**) Representative fluorescence time trace from single-molecule FRET experiments showing number of detected photons for each 100 ms interval as intensity traces (green for donor; red for acceptor) with the average for each period of constant intensity (black solid line) and the corresponding donor lifetime (black dashed line). smFRET donor lifetime distributions with atto647N γATP as acceptor and snap surface 594 as donor in (**j**) neutral (0% POPS) and (**k**) 30% anionic lipids (30% POPS) without EGF (top); with 1 μM EGF (bottom).

The online version of this article includes the following source data and figure supplement(s) for figure 2:

**Source data 1.** PDF file containing original SDS-PAGE gels for **Figure 2b**, indicating the relevant bands and experimental conditions.

**Source data 2.** Original files for SDS-PAGE gels for **Figure 2b**.

**Source data 3.** Raw data underlying **Figure 2b**.

**Source data 4.** Raw data underlying smFRET distribution in **Figure 2j**.

**Source data 5.** Raw data underlying smFRET distribution in **Figure 2k**.

**Figure supplement 1.** Characterization of ss594-labeled EGFR nanodiscs in different anionic lipid environments.

**Figure supplement 2.** Characterization of cy5-labeled EGFR nanodiscs in different anionic lipid environments.

**Figure supplement 3.** Single ss594-labeled EGFR-embedded nanodiscs.

**Figure supplement 4.** Jitter plots of the donor lifetime distributions from smFRET experiments to measure the distance between the ATP binding site and the C-terminus of EGFR.

**Figure supplement 4—source data 1.** Raw data underlying **Figure 2—figure supplement 4**.

interactions with anionic lipids, can regulate accessibility of the ATP binding site, likely as a mechanism to modulate EGFR catalytic activity.

The accessibility of the ATP binding site was also examined through additional molecular dynamics simulations (*Figure 2c–f*). Simulations were carried out on EGFR in a neutral and partially (30%) anionic lipid bilayer in the absence and presence of EGF. For each condition, the number of contacts between the ATP binding site and the membrane was determined (*Figure 2c and d*). The number of contacts was defined as the number of coarse-grained atom pairs between the lipid membrane and the ATP binding site that have a smaller than 16 Å distance. In the absence of EGF, increasing the anionic lipid content from 0% POPS to 30% POPS increased the number of ATP-lipid contacts from 58.6±0.7–74.4±1.2, indicating reduced accessibility, consistent with the experimental results and suggesting anionic lipids are required for ligand-induced EGFR activity. In the presence of EGF, increasing the anionic lipid content decreased the number of contacts from 71.8±1.8–67.8±2.4, indicating increased accessibility, again in line with the experimental findings. Because detection of EGFR relies on labeling at the C-terminus and ATP binding requires an intact kinase domain, the ATP-binding assay is for receptors that are properly folded and competent for nucleotide binding. The consistency between experimental results and MD simulations suggests that the observed lipid-dependent changes are more likely due to modulation of functional EGFR than to artifacts from misfolding.

## Anionic lipid dependence of kinase domain position

To experimentally investigate the intracellular conformations responsible for the lipid-dependent ATP accessibility, we performed smFRET measurements that probed the relative organization of the kinase domain and the C-terminal tail. EGFR was embedded in 0% and 30% POPS nanodiscs with snap surface 594 on the C-terminal tail as the donor and a fluorescently labeled ATP analog (atto647N) as the acceptor (*Figure 2a*). The affinity tag on the belting protein was used for immobilization of EGFR nanodiscs at dilute concentration on a coverslip so the donor fluorescence could be recorded for individual constructs (*Figure 2g–i*, *Figure 2—figure supplement 3*; *Srinivasan et al., 2022*). These measurements monitored the donor fluorescence lifetime, which decreases with donor-acceptor distance or as energy transfer to the acceptor increases (*Sisamakis et al., 2010*). The donor lifetime distributions of the EGFR samples in the absence and presence of EGF (1 $\mu$M) are shown in *Figure 2j and k*, *Table 1*, *Supplementary file 1A and B*. The corresponding donor-acceptor distances were calculated using the lifetime of the donor only sample as a reference (*Figure 2—figure supplement 4*). In the neutral bilayer (0% POPS), the distributions in the absence of EGF peak at 8.1 nm (95% CI: 8.0–8.2 nm) and in the presence of EGF peak at 8.6 nm (95% CI: 8.5–8.7 nm; *Table 1*, *Supplementary file 1A*). In the physiological regime of 30% POPS nanodiscs, the peak of the donor lifetime distribution shifts from 9.1 nm (95% CI: 8.9–9.2 nm) in the absence of EGF to 11.6 nm (95% CI: 11.1–12.6 nm) in the presence of EGF (*Table 1*, *Supplementary file 1A*), which is a larger EGF-induced conformational response than in neutral lipids. The larger conformational response observed in the presence of anionic lipids suggests that these lipids enhance the responsiveness of the intracellular domains to EGF, potentially facilitating interactions between C-terminal sites and adaptor proteins during downstream signaling.

**Table 1.** Median distances between the ATP binding site (residue 721) and the C-terminal end of EGFR from smFRET experiments and simulations.

The distance values were extracted as the medians of the distributions in *Figure 2e, f, j and k* in the main text. The numbers in parentheses indicate the 95% confidence interval for experiments and the minimal and maximal median value from three equal partitions of data for simulations. Ro = 7.5 nm for snap surface 594 and atto 647 N.

| Sample conditions | Experiment distance (nm) | Simulation distance (nm) |
|---|---|---|
| 0% anionic lipids | | |
| EGFR, -EGF | 8.1 [8.0, 8.2] | 7.0 [6.0, 8.0] |
| EGFR,+EGF | 8.6 [8.5, 8.7] | 5.8 [5.2, 7.2] |
| 30% anionic lipids | | |
| EGFR, -EGF | 9.1 [8.9, 9.2] | 5.7 [5.7, 5.7] |
| EGFR,+EGF | 11.6 [11.1, 12.6] | 7.5 [6.2, 9.2] |

The same distance between the C-terminus of the protein and ATP binding site was extracted from the molecular dynamics simulations for both neutral and 30% anionic membranes (*Figure 2c*). In the neutral bilayer, the distance was 7.0 nm and 5.8 nm in the absence and presence of EGF, respectively (*Figure 2e*, *Table 1*). While a small (~1 nm) compaction was observed, both values correspond to an overall similar conformation. Upon introduction of 30% anionic lipids in the bilayer, the measured distance shifted from 5.7 nm in the absence of EGF to 7.5 nm in the presence of EGF (*Figure 2f*, *Table 1*). Both experimental and computational results show a larger EGF-induced shift in the partially anionic bilayer, consistent with the notion that a partially anionic lipid bilayer provides a more native environment that supports proper receptor activation, compared to the non-physiological neutral membrane.

## smFRET measurements of EGFR in membrane nanodiscs

To further map out the conformational response, the overall intracellular conformation of EGFR in the absence and presence of extracellular EGF binding was captured using an additional series of smFRET measurements for different membrane compositions. EGFR nanodiscs containing 0%, 15%, 30%, or 60% of the anionic lipid POPS doped into the POPC bilayer were prepared with snap surface 594 on the C-terminal tail as the donor and the fluorescently-labeled lipid (Cy5) as the acceptor (*Figure 3a*, *Figure 1—figure supplement 2*, *Figure 3—figure supplement 1*). The donor lifetime was measured for all membrane compositions in the presence and absence of saturating concentrations (1 $\mu$M) of EGF. Histograms of the donor lifetimes were constructed for all conditions (*Figure 3*, *Figure 3—figure supplement 2*, *Table 2*, *Supplementary file 1C and D*).

Examination of the distributions showed signatures of a bimodal structure, indicating two-state behavior with a distinct conformational equilibrium for each sample (*Supplementary file 1E*). To quantify the equilibrium for the samples, we globally fit the lifetime distributions for all the samples using maximum likelihood estimation with a double Gaussian distribution model (*Woody et al., 2016*). In global fit, the peak positions and widths are shared parameters between the samples and only the relative amplitudes of the two states change for each distribution. The results of the global fitting are displayed as solid lines in *Figure 3b, d and f*, *Supplementary file 1F*. The global fitting identified peaks at 1.3 ns (~ 8 nm) and 2.7 ns (~ 12 nm) with widths of 0.35 ns (~ 1.9 nm) and 0.67 ns (~ 3 nm), respectively. These two peaks represent a compact and an open conformation, *i.e.*, the C-terminal tail close to and away from the lipid bilayer surface, respectively.

## Anionic lipid dependence of transmembrane conformational response

To investigate the role of anionic lipids in EGFR transmembrane conformational response, the smFRET distributions were compared for nanodisc samples with increasing anionic lipid content (*Figure 3a*, *Table 2*, *Supplementary file 1C and D*). In the physiological regime for anionic lipid content (15–30% POPS; *Figure 3b*), the smFRET distributions exhibited an EGF-induced conformational response. In the absence of EGF, a conformational equilibrium was observed where the amplitude of the open conformation was similar at 69% and 74% for 15% and 30% POPS, respectively (*Figure 3b*, *Supplementary file 1F*). In the presence of EGF, the conformational equilibrium was dominated by the open conformation with an amplitude of 91% and 96% for 15% and 30% POPS, respectively (*Figure 3b*, *Supplementary file 1F*). For both lipid compositions, this corresponds to an increase in amplitude of the open conformation by ~25%. (*Figure 3g and h*). These results suggest that EGF-induced transmembrane conformational response is robust around the physiological anionic lipid content (*Figure 3g and h*, *Supplementary file 1F*).

In the neutral bilayer (0% POPS; *Figure 3b*), the distributions also exhibited an EGF-induced conformational response, but the response was the reverse of that in the physiological regime. As described in the previous section, in the absence of EGF, the conformational equilibrium was dominated by the open conformation at 87%, whereas in the presence of EGF, an equilibrium was observed where the amplitude of the open conformation was 60% (*Figure 3b*, *Supplementary file 1F*). The reversal of the conformational response in the neutral bilayer strongly suggests that electrostatic interactions are playing an important role in transmembrane conformational response, consistent with previous smFRET experiments on EGFR (*Srinivasan et al., 2022*).

In the highly anionic bilayer (60% POPS; *Figure 3b*), the EGF-induced conformational response nearly disappeared. In the absence of EGF, the conformational equilibrium was dominated by the

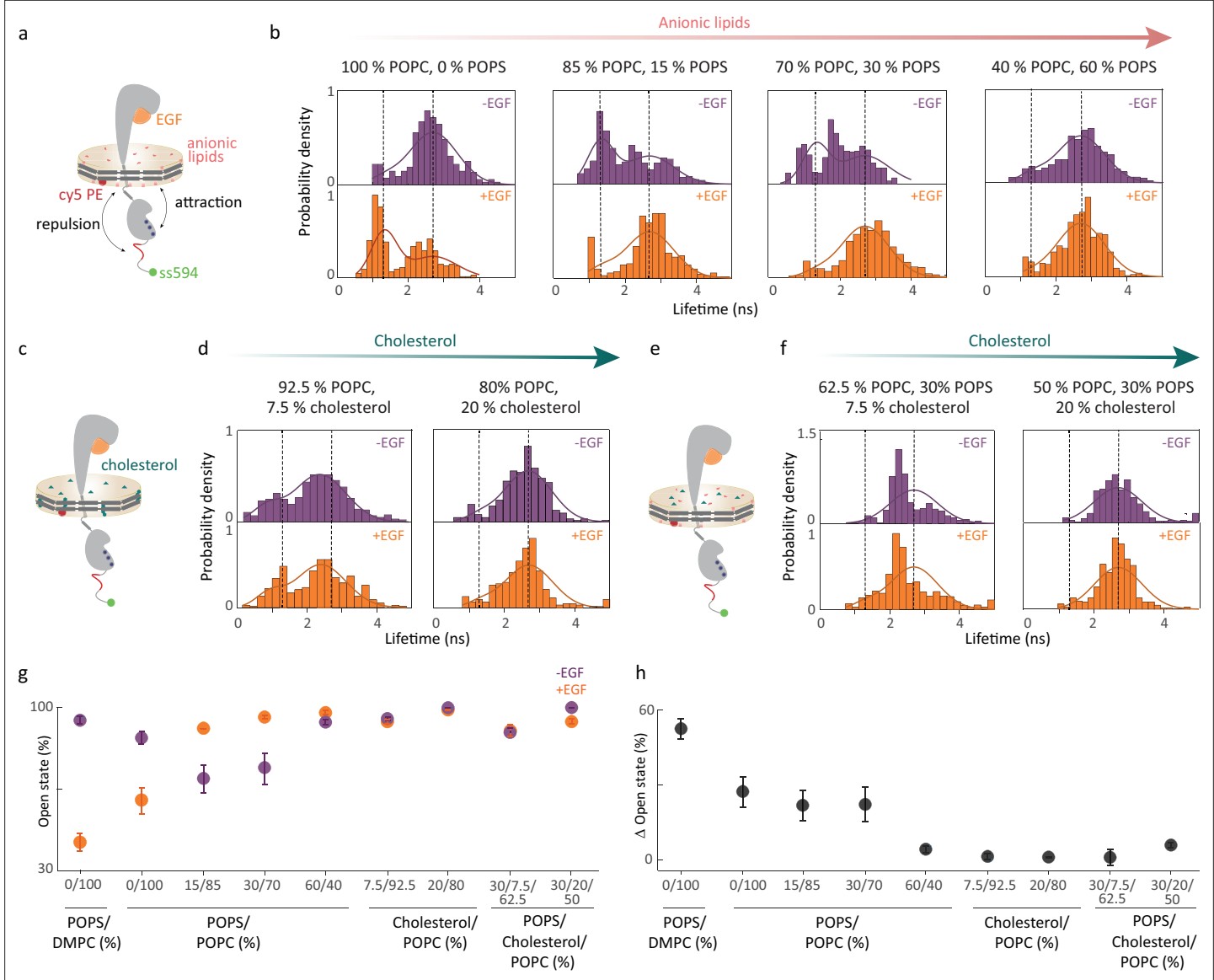

**Figure 3.** EGFR intracellular conformations depend on membrane composition of the nanodisc. (**a**) Full-length EGFR in nanodiscs with partially anionic lipids (red). The negatively charged residues on the C-terminal tail are indicated in red, and the positively charged residues on the kinase domain are indicated in blue. smFRET donor fluorescence lifetime distributions in (**b**) 100% POPC, 0% POPS; 85% POPC, 15% POPS; 70% POPC, 30% POPS; 40% POPC, 60% POPS without EGF (top); with 1 $\mu$M EGF (bottom). (**c**) Full-length EGFR in nanodiscs with cholesterol (teal). (**d**) smFRET donor fluorescence lifetime distributions in 92.5% POPC, 7.5% cholesterol; 80% POPC, 20% cholesterol without EGF (top); with 1 $\mu$M EGF (bottom). (**e**) Full-length EGFR in nanodiscs with cholesterol and anionic lipids. (**f**) smFRET donor fluorescence lifetime distributions in 62.5% POPC, 30% POPS, 7.5% cholesterol; 50% POPC, 30% POPS, 20% cholesterol without EGF (top); with 1 $\mu$M EGF (bottom). Dotted lines indicate the maxima from a global fit of all lifetime distributions to a double Gaussian distribution model using maximum likelihood estimation. The maxima correspond to a compact and an open conformation with a distance of 8 nm and 12 nm, respectively, between the EGFR C-terminal tail and the membrane bilayer. (**g**) The amplitude of the open conformation (in %) in all the eight different membrane compositions in the absence (purple) and presence of EGF (orange). (**h**) The change in amplitude induced by EGF (black). The amplitude change upon EGF addition is high (22–55%) in 0–30% POPS but reduces drastically (0–6%) upon introduction of cholesterol in the lipid bilayer. The error bars in (**g**) and (**h**) are from the global fit.

The online version of this article includes the following source data and figure supplement(s) for figure 3:

**Source data 1.** Raw data underlying smFRET distribution in *Figure 3b*.

**Source data 2.** Raw data underlying smFRET distribution in *Figure 3d*.

**Source data 3.** Raw data underlying smFRET distribution in *Figure 3f*.

**Figure supplement 1.** Single Cy5-labeled EGFR-embedded nanodiscs.

*Figure 3 continued on next page*

*Figure 3 continued*

**Figure supplement 2.** Jitter plots of the donor lifetime distributions from smFRET experiments to measure the distance between the membrane and the C-terminus of EGFR.

**Figure supplement 2—source data 1.** Raw data underlying *Figure 3—figure supplement 2*.

**Figure supplement 3.** EGFR intracellular domain conformation is correlated with ATP binding.

**Figure supplement 4.** Jitter plots of the donor lifetime distributions from smFRET experiments to measure the distance between the membrane and the C-terminus of EGFR.

**Figure supplement 4—source data 1.** Raw data underlying *Figure 3—figure supplement 4*.

**Figure supplement 5.** Jitter plots of the donor lifetime distributions from smFRET experiments to measure the distance between the membrane and the C-terminus of EGFR.

**Figure supplement 5—source data 1.** Raw data underlying *Figure 3—figure supplement 5*.

**Figure supplement 6.** smFRET donor lifetime distributions in DMPC and POPC nanodiscs.

**Figure supplement 6—source data 1.** Raw data underlying *Figure 3—figure supplement 6*.

open conformation >90%, which, in the presence of EGF, increased slightly by ~ 5% (*Figure 3b, g and h*, *Supplementary file 1F*). The dominance of the open conformation in both conditions is similar to the EGF-bound conditions in the physiological regime, indicating that the receptor is held in the active conformation by the anionic bilayer. The amplitude of the open conformation correlated with the ATP binding site accessibility from the experimental and computational studies (*Figure 3—figure supplement 3*), consistent with a picture in which the open conformation enables access by all signaling partners. High levels of EGFR signaling are often present in cancer cells, where anionic lipid content is also high (*Ran et al., 2002*). Thus, the open conformation in the anionic bilayer may be constitutively active - and thereby the biophysical origin of the high signaling that drives cancer cell growth. Similar effects may be playing a role in the observed overactivation of EGFR in neurodegenerative disorders. Anionic lipids are increasingly recognized as biomarkers for neurodegenerative diseases, with EGFR emerging as a dual molecular target for both cancer and Alzheimer's disease (*Choi et al., 2023*). Specifically, tyrosine kinase inhibitors, which are commonly used for cancer treatment, show promise in mitigating Alzheimer's disease by targeting this overactive EGFR signaling pathway (*Mansour et al., 2021*).

## Electrostatic interactions in transmembrane conformational response

The molecular dynamics simulations were further examined to investigate how anionic lipids influence EGFR transmembrane coupling. EGF binding was found to switch the extracellular domain from prostrate on the membrane to upright (*Arkhipov et al., 2013*), which, in turn, switched the transmembrane domain from a tilted to vertical orientation (*Srinivasan et al., 2022*). The orientation switch caused the juxtamembrane domain to move from embedded inside to extended outside the lipid

**Table 2.** Median distances between the membrane and C-terminal end of the protein from smFRET experiments.

The distance values were extracted from the distributions shown in *Figure 3b* in the main text. The numbers in parentheses indicate the 95% confidence interval for experiments. Asterisk (*) indicates distance was beyond the FRET range for the snap surface 594 and Cy5 dye pair (Ro = 8.4 nm; *Sisamakis et al., 2010*).

| Sample conditions | Distance (nm) |
|---|---|
| 0% POPS, 100% POPC; -EGF | 12.1 [11.6, 12.7] |
| 0% POPS, 100% POPC; +EGF | 8.8 [8.2, 9.6] |
| 15% POPS, 85% POPC; -EGF | 9.2 [8.8, 9.6] |
| 15% POPS, 85% POPC; +EGF | 11.9 [11.4, 12.4] |
| 30% POPS, 70% POPC; -EGF | 9.2 [9.1, 9.3] |
| 30% POPS, 70% POPC; +EGF | *13.9 [12.5, 14.4] |
| 60% POPS, 40% POPC; -EGF | 11.4 [11.1, 11.7] |
| 60% POPS, 40% POPC; +EGF | 11.2 [10.9, 11.6] |

bilayer (*Srinivasan et al., 2022*).The simulations showed the position of the juxtamembrane domain influenced two key interactions in the intracellular domain: (1) attraction between the basic residues in the juxtamembrane/kinase domains and the anionic lipids; and (2) repulsion between the acidic residues on the N-terminal portion of the C-terminal tail and the anionic lipids (*Figure 1—figure supplement 1*). The competition between these interactions dictates the overall conformation of the intracellular domain.

In the absence of EGF, the more embedded juxtamembrane domain meant that the attraction between the juxtamembrane/kinase domains and the lipids dominated, inducing a more closed conformation. The dominant closed conformation was consistent with the high compact state percentage observed in the smFRET distributions for 15% POPS (*Figure 3b*). Within this picture of competing electrostatic interactions, the repulsion between the C-terminal tail and the anionic lipids is expected to dominate at high anionic lipid content, likely because the number of negative charges on the C-terminus is greater than the number of positive charges on either the juxtamembrane or kinase domains. Consistently, a higher open state percentage was observed for 30% and 60% POPS. In the presence of EGF, the more extended juxtamembrane domain separated the juxtamembrane/kinase domains and the lipids, decreasing their attraction and leading to a more open conformation again consistent with the higher open state percentage for all partially anionic bilayers.

Oncogenic mutations such as K745E and K757E introduce negative charges into the kinase domain, which likely decrease the attraction between the kinase domain and the anionic lipids to allow the repulsion and thus the open conformation to dominate (*Sueangoen et al., 2020*; *de Biase et al., 2017*). As the open conformation also dominates in the presence of EGF (*Figure 3b and g*), these mutations likely increase signaling levels, consistent with their role in cancer. The electrostatic attraction between the kinase domain and anionic lipids described above was previously found to restrict the access of the tyrosines on the C-terminal tail to the catalytic site in the kinase domain (*McLaughlin et al., 2005*). However, the electrostatic repulsion between the C-terminal tail and the lipids had not been captured due to the absence of the C-terminal tail in the previous work (*Mi et al., 2011*).

## Cholesterol inhibits transmembrane conformational response

To investigate the mechanism behind the suppression of EGFR signaling by cholesterol, smFRET experiments were also performed on nanodisc samples with cholesterol incorporated into the lipid bilayer (*Figure 3c–f*). The smFRET distributions were statistically the same in the presence and absence of EGF for all cholesterol-containing samples (*Figure 3d, f*, *Figure 3—figure supplements 4 and 5*, *Table 3*, *Supplementary file 1G–L*), revealing the suppression of the EGF-induced conformational response.

**Table 3.** Median distances between the membrane and C-terminal end of the protein from smFRET experiments.

The distance values were extracted from the distributions shown in *Figure 3d and f* in the main text. The numbers in parentheses indicate the 95% confidence interval for experiments. Asterisk (*) indicates distance was beyond the FRET range for the snap surface 594 and Cy5 dye pair (Ro = 8.4 nm; *Sisamakis et al., 2010*).

| Sample conditions | Experiment distance (nm) |
| --- | --- |
| 92.5% POPC, 7.5% Cholesterol; -EGF | 12.1 [11.6, 12.6] |
| 92.5% POPC, 7.5% Cholesterol; +EGF | 12.2 [11.5, 13.2] |
| 80% POPC, 20% Cholesterol; -EGF | 11.1 [10.7, 11.5] |
| 80% POPC, 20% Cholesterol; +EGF | 10.9 [10.6, 11.2] |
| 62.5% POPC, 30% POPS, 7.5% Cholesterol, -EGF | 10.4 [10.2, 10.5] |
| 62.5% POPC, 30% POPS, 7.5% Cholesterol, +EGF | 10.3 [10.2, 10.4] |
| 50% POPC, 30% POPS, 20% Cholesterol, -EGF | *12.9 [11.5, 12.3] |
| 50% POPC, 30% POPS, 20% Cholesterol, +EGF | *13.0 [11.5, 12.1] |
| 100% DMPC; -EGF | 11.4 [11.1, 11.6] |
| 100% DMPC; +EGF | 8.1 [8.1, 8.2] |

In the physiological regime of cholesterol content (20%), the distributions were dominated by the open conformation with an amplitude of ~95% (*Figure 3g and h*, *Supplementary file 1I and L*). For low cholesterol content (7.5%), the distributions were still dominated by the open conformation with a slightly smaller amplitude of ~90%, corresponding to a shift in the peak of the distribution to shorter distances (*Supplementary file 1I and L*). Again, the distributions were statistically the same in the presence and absence of EGF (*Figure 3—figure supplements 4 and 5*, *Supplementary file 1G and J*), indicating only a small amount (≤7.5%) of cholesterol is required to suppress the conformational response.

Consistent with the suppression of the conformational response, several previous observations show that cholesterol impedes EGFR function, leading to the hypothesis that it must be sequestered for proper signaling (*Takayama et al., 2024*). An artificial increase in the cholesterol content of the plasma membrane for A431 cells or HEp-2 cells led to reduced ligand-induced EGFR activation (*Ringerike et al., 2002*). Cholesterol depletion by the drugs MβCD or U18666A also increased dimerization and phosphorylation of EGFR (*Ringerike et al., 2002*; *Chen and Resh, 2002*). The introduction of cholesterol into the membrane of proteoliposomes containing reconstituted EGFR resulted in decreased kinase activity, which was attributed to a change in membrane fluidity (*Ge et al., 2001*). Although EGFR is locked in the EGF-bound configuration, the suppressed conformational response may play a role in the biophysical mechanism behind the ability of cholesterol to impede EGFR signaling (*Ringerike et al., 2002*). Remarkably, high anionic lipids and cholesterol content produce the same EGFR conformations but with opposite effects on signaling—suppression or enhancement. Both conditions, however, underscore the complexity and sensitivity of membrane regulation in cell signaling.

## Mechanism of cholesterol inhibition of EGFR transmembrane conformational response

Cholesterol is known to impact both the thickness and the fluidity of lipid bilayers. Previous computational and experimental results showed high cholesterol can increase the thickness of a phospholipid bilayer, defined as the separation between lipid phosphate head groups, by ~ 0.3–0.5 nm (*Prakash et al., 2011*; *Chen et al., 2023*). Evidence suggests that transmembrane helices tilt to minimize hydrophobic mismatch (*Prakash et al., 2011*), changing the conformation of embedded membrane proteins. The incorporation of cholesterol into the lipid bilayer has also been shown to alter its fluidity. The altered fluidity was also found to modulate the mobility and oligomerization of ErbB2, a member of the EGFR family (*Sharpe et al., 2002*). Either factor —bilayer thickness or fluidity —could, therefore, give rise to the suppressed conformational response observed in the smFRET measurements.

To identify the mechanistic origin of the cholesterol-induced suppression of the conformational response, smFRET experiments were performed on EGFR-containing nanodiscs with two different lipids, DMPC and POPC (*Figure 3—figure supplement 6*, *Table 3*, *Supplementary file 1C-F*). DMPC is a fully saturated version of POPC that forms a thinner bilayer (3.7 nm for DMPC and 3.9 nm for POPC *Kučerka et al., 2011*), which is similar in magnitude to the thickness change observed upon the addition of cholesterol (*Prakash et al., 2011*; *Chen et al., 2023*). The fluidity of the lipid bilayer is described through the order parameter ($S$), which is the average ordering of the lipid chains. $S$ is ~ 0.3 for nanodiscs of the same size and containing the same number of DMPC or POPC lipids, indicating comparable fluidity (*Siuda and Tieleman, 2015*).

In both POPC and DMPC bilayers, the distribution was dominated by the open conformation (~12 nm) in the absence of EGF, whereas the equilibrium shifted towards the compact conformation (~ 8 nm) in the presence of EGF (*Figure 3—figure supplement 6*). The amplitude of the open conformation decreased from ~87% in the absence of EGF to 60% in its presence for POPC and decreased from 95% in the absence of EGF to 42% in its presence for DMPC (*Supplementary file 1F*). The similar behavior between POPC and DMPC strongly implies the transmembrane conformational response is independent of bilayer thickness, suggesting that the decreased membrane fluidity in the presence of cholesterol is likely responsible for the suppression of the conformational response.

## Implications of lipid dependence on EGFR and other membrane receptors

EGFR activation has long been understood through the lens of ligand-induced structural rearrangements and downstream protein-protein interactions. However, studies of these components were unable to evaluate the role of the surrounding membrane. Our findings introduce a direct effect of membrane composition on the conformational response of EGFR, explaining previous observations of membrane-mediated regulation of downstream signaling. Here, we report the discovery that the lipid composition—through both electrostatics and membrane mechanics—can override ligand-driven activation to regulate receptor function (*Figure 4*).

Notably, we found that the conformational response is suppressed in membrane compositions known to impede healthy signaling, that is, high cholesterol or high anionic lipid content. While autoinhibitive interactions of lipids with individual EGFR domains had been established previously (*McLaughlin et al., 2005*; *Endres et al., 2013*; *Arkhipov et al., 2013*), our results show that the protein-lipid interplay is crucial to maintain a ligand-induced transmembrane conformational response, which is the key regulator of healthy signaling. Our results complement and expand upon previous molecular dynamic simulations of the kinase domain that increasing amounts of anionic lipids (up to 30%) induce a corresponding decrease in the accessibility of the ATP binding site (*Arkhipov et al., 2013*). Similar interactions were observed for PIP2 lipids as for PS (*Michailidis et al., 2011*; *Abd Halim et al., 2015*; *Wang et al., 2014*; *Srinivasan et al., 2022*), suggesting similar effects may be induced during PIP2-mediated processes. Apart from phospholipids and sterols, glycolipids have been shown to have pronounced effects on EGFR activity (*Coskun et al., 2011*; *Daniotti et al., 2006*). Ganglioside GM3 inhibits cell motility via downregulation of ligand-stimulated EGFR phosphorylation and signaling (*Li et al., 2015*; *Huang et al., 2013*). Our results suggest that these components may function through conformational effects similar to the ones observed here.

In addition to EGFR, membrane composition is crucial for the activity of many membrane proteins (*Phillips et al., 2009*). For example, rhodopsin has an important and functional interaction with G proteins that hinges on the presence of POPS lipids. These lipids are essential for maintaining the structure of its key amphipathic helix, similar to the POPS dependence observed here (*Palczewski et al., 2000*; *Krishna et al., 2002*). Appropriate lipid size is also necessary to match membrane thickness with transmembrane regions of integral proteins, preventing hydrophobic mismatches that could cause structural anomalies or significantly impair protein function (*Killian, 1998*; *de Planque and Killian, 2003*), such as 80% reduction in $Ca^{2+}$-ATPase activity (*Pilot et al., 2001*; *Lee, 2003*). Furthermore, the saturation level of lipid hydrocarbon tails, influencing their melting temperature, is critical in determining membrane bilayer states and protein activity (*Cornelius, 2001*). Thus, the insights into the membrane dependence of EGFR signaling reported here could be universal to other membrane receptors, particularly the family of proteins that share the same structural homology and perform other significant functions and suggest that the membrane has broad implications in cellular signaling.

## Methods

### Production of labeled full-length EGFR nanodiscs

Fluorescently-labeled EGFR in nanodiscs was produced and characterized as described in previously published protocols (*He et al., 2015*; *Quinn et al., 2019*; *Srinivasan et al., 2022*). ApoA1Δ49 in pIVEX2.4d vector (Genscript) and full-length EGFR (1210 amino acids) with SNAP tag at the C-terminus in SNAP T7 vector (Genscript) was added to the cell-free reaction contents (Thermo Fisher Scientific; *Figure 1—figure supplement 2*). Briefly, the *E. coli* slyD lysate, in vitro protein synthesis *E. coli* reaction buffer, amino acids (-Methionine), Methionine, T7 Enzyme, protease inhibitor cocktail (Thermo Fisher Scientific), RNAse inhibitor (Roche), and DNA plasmids (20 μg of EGFR and 0.2 μg of ApoA1Δ49) were mixed with different lipid mixtures. The DNA template ratio of EGFR:ApoA1Δ49=100:1 was empirically chosen by testing different ratios on SDS-PAGE gels and selecting the condition that maximized full-length EGFR expression in DMPC lipids (*Figure 1—figure supplement 3*). The lipid mixtures were made by sonicating different percentages of 1,2-dimyristoyl-sn-glycero-3-phosphoholine (DMPC; Avanti Polar Lipids), 1-palmitoyl-2-oleoyl-sn-glycero-3-phosphocholine (POPC; Avanti Polar Lipids), 1-palmitoyl-2-oleoyl-sn-glycero-3-phospho-L-serine (POPS; Avanti Polar Lipids) and cholesterol (Sigma-Aldrich) keeping the total lipid concentration at 2 mg/mL for the cell-free reaction

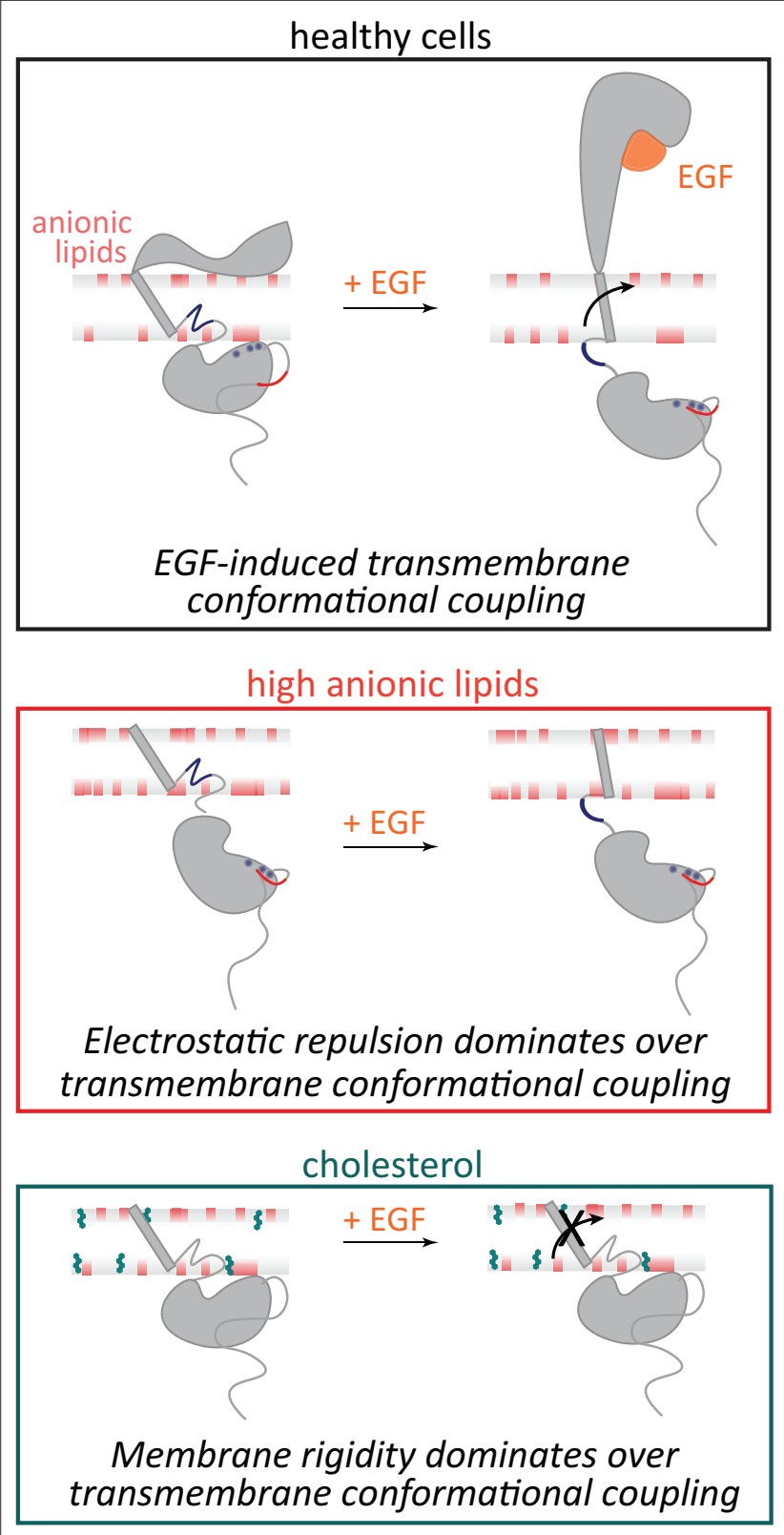

**Figure 4.** Membrane composition influences EGFR conformation and function. (Top) In healthy cells, EGF binding promotes transmembrane conformational coupling between the extracellular and intracellular domains of EGFR, enabling kinase activation through established signaling mechanisms. (Middle) Our results demonstrate that in membranes with high anionic lipid content, electrostatic repulsion between the negatively charged lipids and the

*Figure 4 continued on next page*

*Figure 4 continued*

kinase domain overrides EGF-induced transmembrane conformational coupling. (Bottom) Our results demonstrate that in cholesterol-rich membranes, increased membrane rigidity overrides EGF-induced transmembrane conformational coupling. These findings establish membrane composition as a dominant regulator of EGFR signaling, independent of EGF binding.

mixture. For introducing a single acceptor into the nanodisc, a molar ratio of 500:1 lipid:cy5 labeled,2-dioleoyl-sn-glycero-3-phosphoethanolamine (DOPE) lipids (Avanti Polar lipids) was added to the above solution after bath sonication. The above solution was incubated at 25 °C, 300 rpm for 30 min. The mixture was supplemented with *E. coli* feed buffer, amino acids (-Methionine), and Methionine (total volume of cell-free reaction is 250 µL) and incubated for additional 8 hr. 500 nM of snap surface 594 (New England Biolabs) was next added to the above mixture and incubated at 37 °C, 300 rpm for 35 min. Snap surface 594, a derivative of Atto594 with benzylguanine functionality, reacts in a near-stoichiometric efficiency with genetically encoded snap tag (*Sun et al., 2011*).

### Affinity purification of labeled EGFR nanodiscs

500 µL of Ni-NTA resin slurry (Qiagen) was added to a 2 mL plastic column (Bio-Rad Laboratories). The resin was washed with double-distilled water and equilibrated with 3 mL of native lysis buffer (50 mM $NaH_2PO_4$, 300 mM NaCl, pH 8.0). The cell-free reaction post-labeling was added to 500 µL of lysis buffer on the incubated column and incubated at 4 °C for 2 hr. After the flowthrough was collected, the column was washed with lysis buffer (3×1 mL) followed by lysis buffer containing 10 mM imidazole (2×1 mL) and lysis buffer containing 25 mM imidazole (2×1 mL) to remove all the non-specific interactions of the reaction mixture and free dye from the column. The EGFR nanodiscs were eluted with lysis buffer containing 400 mM (2×500 µL) imidazole. The eluted fractions were concentrated using 50 kDa, 500 µL spin filters (Sigma-Aldrich) by centrifugation.

### Protein content for labeled EGFR nanodiscs

SDS-PAGE was used to confirm the production of both belt protein (at 25 kDa) and EGFR (at 160 kDa). Samples were mixed with 2×Laemmli sample buffer (Bio-Rad Laboratories), 2.5% 2-mercaptoethanol (Sigma-Aldrich), and boiled for 5 min at 95 °C before running on precast stain-free gels from Bio-Rad Laboratories. Precision Plus Protein unstained standard (Bio-Rad Laboratories) marker was used for the stain-free imaging and Prestained NIR protein ladder (Thermo Fisher Scientific) for fluorescence imaging. Gels were run at 170 V for 45 min. Stain-free and fluorescent imaging was performed on ChemiDoc Imaging System (Bio-Rad Laboratories). The other proteins appearing in the stain-free gel are proteins not expressed completely during the cell-free reaction or the transcription and translation machinery of the cell-free reaction. The specificity of snap surface 594 fluorophore binding to EGFR was confirmed through the fluorescence gel.

### Transmission electron microscopy

5 µL of cell-free expressed EGFR nanodiscs in 1×PBS buffer (137 mM NaCl, 2.7 mM KCl, 10 mM $Na_2HPO_4$, 1.8 mM $NaH_2PO_4$, pH 7.4) was added to glow-discharged carbon coated 400 mesh copper grids (Electron Microscopy Sciences) and incubated for 5 min at room temperature to allow nonspecific binding of the nanodiscs to the grids. The solutions were removed by gently blotting the side of the grid with filter paper. The grids were subsequently incubated with 5 µL of 2% aqueous uranyl acetate for 30 s. Excess stain was removed similarly to the sample. The grids were air-dried and then imaged on a FEI Tecnai transmission electron microscope (120 kV, 0.35 nm point resolution). The distribution of disc sizes was analyzed using Image J software.

### Dynamic light scattering

The EGFR nanodiscs in 1×PBS buffer were filtered using 0.2 µm syringe filters and their dynamic light scattering measurements were performed on a DynaPro NanoStar (Wyatt Technologies, USA). Each measurement represents an average of 50 individual runs. Substantial heterogeneity in nanodisc lipid composition, such as uneven incorporation of cholesterol, would be expected to broaden the DLS distributions. However, comparison of the full width at half maximum (FWHM) values from the DLS

distributions showed no significant broadening between cholesterol-containing and cholesterol-free nanodiscs (Mann-Whitney U test, p=0.486; n=4 for each group).

## Zeta potential measurements to quantify surface charge of nanodiscs

Titrations (0 %, 15 %, 30%, and 60 %) of negatively charged POPS lipids in neutral POPC lipids were performed to determine the surface charge of the nanodiscs with increasing negatively charged lipid content. Zeta potential measurements were performed on a Malvern Zetasizer Nano – ZS90 (Malvern, UK), with a backscattering detection at a constant 173° scattering angle, equipped with a laser at 4 mW, 633 nm. Dip cell ZEN1002 (Malvern UK) was used in the zeta-potential experiments. EGFR-loaded nanodiscs produced from cell-free reactions were purified as mentioned above and buffer exchanged to 0.1×PBS. A final volume of 650 uL was prepared and transferred into the zeta dip cell. For each sample, a total of 5 scans, 30 runs each, with an initial equilibration time of 5 min, were recorded. All experiments were performed at 25 °C. Values of the viscosity and refractive index were set at 0.8878 cP and 1.330, respectively. Data analysis was processed using the instrumental Malvern's DTS software to obtain the mean zeta-potential value. This ensemble measurement reports the average surface charge of the nanodisc population, verifying incorporation of anionic POPS lipids.

## Fluorescence measurements with Laurdan to confirm cholesterol insertion into nanodiscs

Vesicles were prepared by mixing phospholipids and cholesterol in the desired ratios in chloroform such that the total concentration of the lipids was 20 mg/mL. Laurdan was added in 100:1 lipid:laurdan ratio. The solvent was first evaporated by nitrogen flow, followed by overnight drying in a vacuum desiccator. The dried samples were resuspended in water such that the total concentration of the lipids was 20 mg/mL. The samples were then heated to 70 °C and vortexed for 5–6 hr. After vortexing, the lipids were used in the cell-free reaction as described above. The EGFR nanodiscs containing laurdan were purified as described above and diluted with 1 X PBS. The fluorescence excitation and emission spectra were recorded for the EGFR nanodiscs containing 0%, 7.5%, and 20% cholesterol in the nanodiscs in Cary Eclipse fluorescence spectrophotometer (Agilent). The excitation spectrum was recorded by collecting the emission at 440 nm and the emission spectrum was recorded by exciting the sample at 385 nm. Laurdan fluorescence provides an ensemble readout of membrane order and confirms cholesterol incorporation into the nanodisc population. While laurdan does not resolve the composition of individual nanodiscs, prior work has shown that POPC-cholesterol mixtures are miscible without forming cholesterol-rich domains (*Veatch and Keller, 2003*; *Risselada and Marrink, 2008*), thus the observed ordering changes likely reflect the intended input cholesterol content at the ensemble level.

## Phosphorylation of EGFR in nanodiscs

Western blot was performed using the Trans-Blot Turbo Transfer System (Bio-Rad Laboratories). The pre-loaded program for high-molecular-weight protein transfer was used for membrane transfer. After the transfer, the membrane was blocked in 5% Non-Fat Dry Milk (prepared in TBST buffer) for the anti-EGFR western blots, and in 5% BSA (prepared in TBST buffer) for the anti-phosphotyrosine Western blots for 20 min at room temperature. The membrane was then incubated in primary antibody over-night at 4 °C. The following day, the membrane was washed and incubated with secondary antibody for 1 hr at room temperature. The primary antibodies, secondary antibodies, and dilutions are listed in *Supplementary file 1M*. The fluorescence band detection was done using the ChemiDoc Imaging System (Bio-Rad Laboratories).

## Preparation of EGF ligand

Human EGF produced in *E. coli* was purchased from Gold Biotechnology (Catalog Number: 1150-04-100). 1 $\mu$M EGF was prepared in 1×PBS buffer.

## ATP binding experiments

Full-length EGFR in different lipid environments was prepared using cell-free expression as described above. 1 $\mu$M of snap surface 488 (New England Biolabs) and atto647N labeled gamma ATP (Jena Bioscience) was added after cell-free expression reaction and incubated at 30°, 300 rpm for 60 min.

1 $\mu$M of atto647N gamma ATP was used, corresponding to a concentration near the reported $K_m$ of 5.2 $\mu$M for ATP binding to the isolated EGFR kinase domain (**Yun et al., 2008**), ensuring sensitivity to lipid-dependent changes in ATP accessibility. Purification using Ni-NTA affinity was performed and the samples were concentrated as described above. The samples were run on a SDS-PAGE gel at 170 V for 40 min and imaged using the ChemiDoc Imaging System (Bio-Rad Laboratories).

## Fluorescence spectroscopy

The His-tag present on the belt protein (ApoA1Δ49) was used to immobilize the EGFR-nanodisc constructs onto the microscope coverslip *via* Ni-NTA affinity. The purified EGFR nanodiscs were diluted to ~500 pM in 1×PBS buffer and incubated for 15 min on the Ni-NTA-coated glass (from Microsurfaces, Inc) and flushed with solution containing 2 mM 6-hydroxy-2,5,7,8-tetramethylchroman-2-carboxylic acid (Sigma-Aldrich), 25 nM protocatechuate-3,4-dioxygenase (Sigma-Aldrich), and 2.5 mM protocatechuic acid (Sigma-Aldrich). Fluorescence experiments were then carried out on a home-built confocal microscope (**Kondo et al., 2017**). A Ti-Sapphire laser (Vitara-S, Coherent: $\lambda_c$ = 800 nm, 70 nm bandwidth, 20 fs pulse duration, 80 MHz repetition rate) was focused into a non-linear photonic crystal fiber (FemtoWhite 800, NKT Photonics) to generate a supercontinuum. Excitation light was then spectrally filtered for pulses centered at 550 nm or 640 nm and focused with an oil immersion objective lens (UPLSAPO100×, Olympus, NA = 1.4). Fluorescence emission was collected by the same objective and fed to the avalanche photodiodes (SPCMAQRH-15, Excelitas). A 5 $\mu$m ×5 $\mu$m area of a coverslip with immobilized receptors was scanned. Diffraction-limited and spatially separated single molecule spots were then probed individually by unblocking the laser beam to record fluorescence until photo-bleaching. For 550 nm excitation, fluorescence was separated with a dichroic filter SP01-561RU (Laser 2000) and passed through FF01-629/56-25 (Semrock) for donor fluorescence collection. For experiments at 640 nm, ET 645/30× (Chroma) was used as the excitation filter, FF01-629/56-25 (Semrock) as the dichroic and FF02-685/40-25 (Semrock) for acceptor fluorescence collection. The laser power for the experiments was 2–3 $\mu$W at the sample plane.

Florescence emission was binned at 100 ms resolution to generate fluorescence intensity traces for both the donor and acceptor channels. Traces with a single photobleaching step for the donor and acceptor were considered for further analysis. Regions of constant intensity in the traces were identified by a change-point algorithm (**Watkins and Yang, 2005**). Donor traces were assigned as FRET levels until acceptor photobleaching. The presence of empty nanodiscs does not influence these measurements, as photobleaching and single-molecule FRET analyses selectively report on receptor-containing nanodiscs. Consecutive bunches of 1000 photons in the donor channel were used to construct fluorescence decay curves for the FRET levels (**Regmi et al., 2020**). The photons were histogrammed and the distributions were fit to a mono-exponential function convolved with the instrument response function (IRF) and summed with a separately-measured background term. The fit was performed using a Maximum Likelihood Estimator (MLE), which has been shown to be more accurate in the single-molecule regime (**Kondo et al., 2017**; **Goldsmith and Moerner, 2010**). The extracted lifetimes were used to construct histograms with bin sizes estimated from the square root of the total number of photon bunches.

The donor-acceptor distance (r in nm) was estimated using the following relation (**Sisamakis et al., 2010**; **Medintz and Hildebrandt, 2013**):

$$r = r_o \sqrt[6]{\frac{1-E}{E}} \tag{1}$$

where $r_o$ is the calculated Förster distance (8.4 nm for snap surface 594 and cy5 dye pair; 7.5 nm for snap surface 594 and atto 647 N dye pair; **Sisamakis et al., 2010**) and the FRET efficiency (E) being experimentally measured as:

$$E = 1 - \frac{\tau_{DA}}{\tau_D} \tag{2}$$

$\tau_{DA}$ is the fluorescence lifetime of the donor in the presence of an acceptor, and $\tau_D$ is the lifetime of donor-only construct. The distance between the donor and acceptor was quantified using a reference lifetime determined with a separately characterized donor-only construct (**Figure 2—figure supplement 4**, **Figure 3—figure supplements 2; 4–6**). For the smFRET measurements on the labeled constructs, the Cy5-lipid is confined to one side of the membrane for the duration of the measurement

(*Kiessling et al., 2006*), and so the extracted distances are an average over the rapid translational diffusion of the labeled dye across the surface of the membrane nanodisc (*Hsieh et al., 2014*).

## Model selection and statistical analysis

Global fitting of lifetime distributions was performed across all experimental conditions using maximum likelihood estimation. Both two-Gaussian and three-Gaussian distribution models were evaluated as described previously (*Woody et al., 2016*). Model performance was compared using the Bayesian Information Criterion (BIC; *Schwarz, 1978*), which balances model likelihood and complexity according to

$$BIC = -2 \ln L + k \ln n \tag{3}$$

where L is the likelihood, k the number of free parameters, and n the number of single-molecule photon bunches across all experimental conditions. A lower BIC value indicates a statistically better model (*Schwarz, 1978*). The separation between Gaussian components was assessed using Ashman's D, where a score above 2 indicates good separation (*Ashman et al., 1994*). For two Gaussian components with means μ1, μ2 and standard deviations σ1, σ2,

$$D_{ij} = \frac{|\mu_i - \mu_j|}{\sqrt{\frac{1}{2}(\sigma_i^2 + \sigma_j^2)}} \tag{4}$$

where $D_{ij}$ represents the distance metric between Gaussian components $i$ and $j$. All fitted parameters, likelihood values, BIC scores, and Ashman's D values are summarized in *Supplementary file 1E*.

## Statistical information

Statistical analysis was performed using MATLAB. One-way analysis of variance (ANOVA) was performed on different pairs of experimental data and statistical significance was set at $p \leq 0.001$. The p-values, degrees of freedom, and F-statistics are reported in *Supplementary file 1ACG and J*. The number of data points in the smFRET lifetime distributions is reported in *Supplementary file 1BDH and K*. The number of data points ranges from two to four biological replicates for each sample.

## Coarse-grained, explicit-solvent simulations with the MARTINI force field

We conducted a series of analyses based on comprehensive explicit-solvent simulations of the full-length EGFR with the coarse-grained MARTINI force field (*de Jong et al., 2013*). Since the original Martini force field (*de Jong et al., 2013*) was parameterized for ordered proteins and will over-collapse the disordered C-terminal tail of EGFR, we calibrated the force field by adjusting the protein-water interactions to capture more accurately the size of the C-terminal tail region (*Srinivasan et al., 2022*). Simulations were performed with the homology-modeled (*Fiser et al., 2000*) full EGFR embedded in a lipid bilayer (*Jo et al., 2008*; *Qi et al., 2015*). 400 $\mu$s, including four sets of simulations, were carried out for active/inactive EGFR embedded in the DMPC/POPC-POPS membranes, respectively. Following the typical Martini protocol (*Qi et al., 2015*), we used a time step of 20 fs, and a temperature of 303 K in all of our simulations. We further used the umbrella sampling technique (*Torrie and Valleau, 1977*) to enhance the exploration of the EGFR conformational space, biasing two collective variables: the contact number between the KD domain and the C-terminal tail, and the distance between the JMA domain and the N-terminal part of the CTT domain. Details of the simulation are described in a previous publication (*Srinivasan et al., 2022*).

We analyzed our simulations using WHAM (*Kumar et al., 1992*; *Noel et al., 2016*) to reweight the umbrella biases and compute the average values of various metrics introduced in this manuscript. Specifically, we calculated the distance between Residue 721 and Residue 1186 (EGFR C-terminus) of the protein. To quantify the accessibility of the ATP-binding site, we calculated the number of contacts between lipid molecules and the residues forming the ATP-binding pocket (residues 694–703, 719, 766–769, 772–773, 817, 820, and 831; *Minnelli et al., 2020*). Close contact between the bilayer and these residues would sterically hinder ATP binding; thus, the contact number serves as a proxy for ATP-site accessibility. The cutoff distance for defining a contact was set to 16 Å, corresponding to the largest molecular radius of the fluorescent ATP analogue

(atto647N-γ ATP, 16.96 Å *Cousins, 2005*). Accordingly, we defined a contact as a pair of coarse-grained atoms, one from the lipid membrane and one from the ATP binding site, within a mutual distance of less than 16 Å.

## Acknowledgements

This work was supported by the NIH MIRA R35 GM148287-02 (to GSS-C). GSS-C also acknowledges a Camille Dreyfus Teacher-Scholar Award. XL and BZ acknowledge support by startup funds from the Department of Chemistry at the Massachusetts Institute of Technology. We thank Dr. Adam W Smith for his valuable feedback on the manuscript.

## Additional information

### Competing interests

Bin Zhang: Reviewing editor, *eLife*. The other authors declare that no competing interests exist.

### Funding

| Funder | Grant reference number | Author |
|---|---|---|
| National Institutes of Health | R35GM148287-02 | Gabriela S Schlau-Cohen |

The funders had no role in study design, data collection and interpretation, or the decision to submit the work for publication.

### Author contributions

Shwetha Srinivasan, Conceptualization, Data curation, Formal analysis, Investigation, Visualization, Methodology, Writing – original draft, Writing – review and editing; Xingcheng Lin, Software, Supervision, Methodology, Writing – review and editing; Xuyan Chen, Formal analysis, Investigation, Methodology; Raju Regmi, Investigation, Methodology, Writing – review and editing; Bin Zhang, Software, Supervision, Funding acquisition, Methodology, Writing – review and editing; Gabriela S Schlau-Cohen, Conceptualization, Supervision, Funding acquisition, Project administration, Writing – review and editing

### Author ORCIDs

Shwetha Srinivasan ⬡ https://orcid.org/0000-0002-8647-6784
Xingcheng Lin ⬡ https://orcid.org/0000-0002-9378-6174
Raju Regmi ⬡ https://orcid.org/0000-0003-4035-0390
Bin Zhang ⬡ https://orcid.org/0000-0002-3685-7503
Gabriela S Schlau-Cohen ⬡ https://orcid.org/0000-0001-7746-2981

Reviewer #1 (Public review): https://doi.org/10.7554/eLife.108789.3.sa1
Reviewer #2 (Public review): https://doi.org/10.7554/eLife.108789.3.sa2
Author response https://doi.org/10.7554/eLife.108789.3.sa3

## Additional files

### Supplementary files

Supplementary file 1. Statistical analyses, sample sizes, model selection, and supporting data for single-molecule FRET measurements across membrane compositions and experimental conditions.

MDAR checklist

### Data availability

All data generated or analyzed during this study are included in the manuscript and supporting files; source data files have been provided for all figures.

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
