## [Editor Report · eLife Assessment]

The authors describe an interesting approach to studying the dynamics and function of membrane proteins in different lipid environments. The **fundamental** findings have theoretical and practical implications beyond the study of EGFR to all membrane signalling proteins. The evidence supporting the conclusions is **compelling**, based on the use of a nanodisk system to study membrane proteins in vitro, combined with state-of-the-art single-molecule FRET. The work will be of broad interest to cell biologists and biochemists.

---

## [Referee Report · Reviewer #1 (Public review)]

Summary:

This work addresses a key question in cell signalling, how does the membrane composition affect the behaviour of a membrane signalling protein? Understanding this is important, not just to understand basic biological function but because membrane composition is highly altered in diseases such as cancer and neurodegenerative disease. Although parts of this question have been addressed on fragments of the target membrane protein, EGFR, used here, Srinivasan et al. harness a unique tool, membrane nanodisks, which allow them to probe full length EGFR in vitro in great detail with cutting-edge fluorescent tools. They find interested impacts on EGFR conformation in differently charged and fluid membranes, explaining previously identified signalling phenotypes.

Strengths:

The nanodisk system enables full length EGFR to be studied in vitro and in a membrane with varying lipid and cholesterol concentrations. The authors combine this with single-molecule FRET utilising multiple pairs of fluorophores at different places on the protein to probe different conformational changes in response to EGF binding under different anionic lipid and cholesterol concentrations. They further support their findings using molecular dynamics simulations which help uncover the full atomistic detail of the conformations they observe.

Weaknesses:

Much of the interpretation of the results comes down to a bimodal model of an 'open' and 'closed' state between the intracellular tail of the protein and the membrane. Some of the data looks like a bimodal model is appropriate but not all. The authors have just this bimodal model statistically and although adding a third component is a better fit, I agree with the authors that it cannot be justified statistically, given the data. Further work beyond the scope of this study would be needed to try to define further states.

---

## [Referee Report · Reviewer #2 (Public review)]

Summary:

Nanodiscs and synthesized EGFR are co-assembled directly in cell-free reactions. Nanodiscs containing membranes with different lipid compositions are obtained by providing liposomes with corresponding lipid mixtures in the reaction. The authors focus on the effects of lipid charge and fluidity on EGFR activity.

Strengths:

The authors implement a variety of complementary techniques to analyze data and to verify results. They further provide a new pipeline to study lipid effects on membrane protein function. The manuscript describes a comprehensive study on the analysis of membrane protein function in context of different lipid environments.

Weaknesses:

As the implemented strategy is relatively new, some uncertainties in the interpretation of the data consequently remain. However, using state-of-the-art techniques, the authors support their results by appropriate data and sufficient controls in the revised manuscript.

---

## [Author Response]

The following is the authors’ response to the original reviews

**Public Reviews:**

**Reviewer #1 (Public review):**
Summary:This work addresses a key question in cell signalling: how does the membrane composition affect the behaviour of a membrane signalling protein? Understanding this is important, not just to understand basic biological function but because membrane composition is highly altered in diseases such as cancer and neurodegenerative disease. Although parts of this question have been addressed on fragments of the target membrane protein, EGFR, used here, Srinivasan et al. harness a unique tool, membrane nanodisks, which allow them to probe full-length EGFR in vitro in great detail with cutting-edge fluorescent tools. They find interesting impacts on EGFR conformation in differently charged and fluid membranes, explaining previously identified signalling phenotypes.Strengths:The nanodisk system enables full-length EGFR to be studied in vitro and in a membrane with varying lipid and cholesterol concentrations. The authors combine this with single-molecule FRET utilising multiple pairs of fluorophores at different places on the protein to probe different conformational changes in response to EGF binding under different anionic lipid and cholesterol concentrations. They further support their findings using molecular dynamics simulations, which help uncover the full atomistic detail of the conformations they observe.Weaknesses:Much of the interpretation of the results comes down to a bimodal model of an 'open' and 'closed' state between the intracellular tail of the protein and the membrane. Some of the data looks like a bimodal model is appropriate, but its use is not sufficiently justified (statistically or otherwise) in this work in its current form. The experiments with varying cholesterol in particular appear to suggest an alternate model with longer fluorescent lifetimes. More justification of these interpretations of the central experiment of this work would strengthen the paper.

We thank the reviewer for highlighting the strengths of the study, including the use of nanodiscs, single-molecule FRET, and MD simulations to probe full-length EGFR in controlled membrane environments.

We agree that statistical justification is important for interpreting the distributions. To address this, we performed global fits of the data with both two- and three-Gaussian models and evaluated them using the Bayesian Information Criterion (BIC), which balances the model fit with a penalty for additional parameters. The three-Gaussian model gave a substantially lower BIC, indicating statistical preference for the more complex model. However, we also assessed the separability of the Gaussian components using Ashman’s D, which quantifies whether peaks are distinct. This analysis showed that two Gaussians (µ = 2.64 and 3.43 ns) are not separable, implying they represent one broad distribution rather than two states.

**Author response table 1. sa3table1:** 

Two-Gaussian		Three-Gaussian	
µ1	1.33 ns	µ1	1.38 ns
σ1	0.34 ns	σ1	0.36 ns
µ2	2.71 ns	µ2	2.64 ns
σ2	0.67 ns	σ2	0.45 ns
		µ3	3.43 ns
		σ3	0.64 ns

Both the two- and three-Gaussian models include a low-value component (µ = ~1.3 ns), but the apparent improvement of the three-Gaussian model arises only from splitting the central population into two overlapping Gaussians. Thus, while the BIC favors the three-Gaussian model statistically, Ashman’s D demonstrates that the central peak should not be interpreted as bimodal. Therefore, when all the distributions are fit globally, the data are best explained as two Gaussians, one centered at ~1.3 ns and the other at ~2.7 ns, with cholesterol-dependent shifts reflecting changes in the distribution of this population rather than the emergence of a separate state. Finally, we acknowledge that additional conformations may exist, but based on this analysis a bimodal model describes the populations captured in our data and so we limit ourselves to this simplest framework.

We have clarified this in the revised manuscript by adding a section in the Methods (page 26) titled Model Selection and Statistical Analysis, which describes the results of the global two- versus three-Gaussian fits evaluated using BIC and Ashman’s D. Additional details of these analyses are also provided in response to Reviewer #1, Question 8 (Recommendations for the authors).

**Reviewer #2 (Public review):**
Summary:Nanodiscs and synthesized EGFR are co-assembled directly in cell-free reactions. Nanodiscs containing membranes with different lipid compositions are obtained by providing liposomes with corresponding lipid mixtures in the reaction. The authors focus on the effects of lipid charge and fluidity on EGFR activity.Strengths:The authors implement a variety of complementary techniques to analyze data and to verify results. They further provide a new pipeline to study lipid effects on membrane protein function.

We thank the reviewer for noting the strengths of our approach, particularly the use of complementary techniques and the development of a new pipeline to study lipid effects on membrane protein function.

Weaknesses:Due to the relative novelty of the approach, a number of concerns remain.(1) I am a little skeptical about the good correlation of the nanodisc compositions with the liposome compositions. I would rather have expected a kind of clustering of individual lipid types in the liposome membrane, in particular of cholesterol. This should then result in an uneven distribution upon nanodisc assembly, i.e., in a notable variation of lipid composition in the individual nanodiscs. Could this be ruled out by the implemented assays, or can just the overall lipid composition of the complete nanodisc fraction be analyzed?

We monitored insertion of anionic lipids into nanodiscs by performing zeta potential measurements, which report on surface charge, and cholesterol insertion by Laurdan fluorescence, which reports on membrane order. Both assays provide information at the ensemble level, not single-nanodisc resolution. We clarified this in the Methods section (see below).

Cholesterol clustering is well documented in ternary systems with saturated lipids and sphingolipids [Veatch, Biophys J., 2003; Risselada, PNAS, 2008]. However, in unsaturated POPC-cholesterol mixtures such as those used here, cholesterol primarily alters bilayer order and large-scale segregation is not typically observed. The addition of POPS to the POPC-cholesterol mixture perturbs cholesterol-induced ordering, lowering the likelihood of cholesterol-rich domains [Kumar, J. Mol. Graphics Modell., 2021].

Lipid heterogeneity between nanodiscs would be expected to give rise to heterogeneity in hydrodynamic properties, including potentially broadening the dynamic light scattering (DLS) distributions. However, the full width at half maximum (FWHM) values from the DLS measurements (see Author response table 2) do not indicate a broadening with cholesterol. Statistical testing (Mann-Whitney U test for non-normal data) showed no significant difference between samples with and without cholesterol (p = 0.486; n = 4 per group). While the sample size is small making firm conclusions challenging, these results suggest that large-scale heterogeneity is unlikely.

**Author response table 2. sa3table2:** 

Sample	FWHM of DLS distribution (nm)
100 % POPC	10
85 %POPC, 15 % POPS	19.1
70 %POPC, 30 % POPS	8.2
40 %POPC, 60 % POPS	18.6
92.5 %POPC, 7.5 % Cholesterol	8.7
80 %POPC, 20 % Cholesterol	21.6
62.5 %POPC, 30% POPS, 7.5 % Cholesterol	23.3
50 %POPC, 30% POPS, 20 % Cholesterol	12.5

In the case of POPS lipids, clustering of POPS in EGFR embedded nanodiscs is a recognized property of receptor-lipid interactions. Molecular dynamics simulations have shown that POPS, although constituting only 30% of the inner leaflet, accounts for ~50% of the lipids directly contacting EGFR [Arkhipov, Cell, 2013], underscoring that anionic lipids are preferentially recruited to the receptor’s immediate environment.

For nanodiscs containing cholesterol and anionic lipids, our smFRET experiments were designed to isolate the effect of EGF binding. The nanodisc population is the same in the ± EGF conditions as EGF was introduced just prior to performing sm-FRET experiments, and not during nanodisc assembly. Thus, for a given lipid composition, any observed differences between ligand-free and ligand-bound states reflect conformational changes of EGFR.

Methods, page 23, “Zeta potential measurements to quantify surface charge of nanodiscs: Data analysis was processed using the instrumental Malvern’s DTS software to obtain the mean zeta-potential value. This ensemble measurement reports the average surface charge of the nanodisc population, verifying incorporation of anionic POPS lipids.”

Methods, page 23, “Fluorescence measurements with Laurdan to confirm cholesterol insertion into nanodiscs: The excitation spectrum was recorded by collecting the emission at 440 nm and emission spectra was recorded by exciting the sample at 385 nm. Laurdan fluorescence provides an ensemble readout of membrane order and confirms cholesterol incorporation into the nanodisc population. While laurdan does not resolve the composition of individual nanodiscs, prior work has shown that POPC–cholesterol mixtures are miscible without forming cholesterol-rich domains[91,92], thus the observed ordering changes likely reflect the intended input cholesterol content at the ensemble level.”

(91) Veatch, S. L. & Keller, S. L. Separation of liquid phases in giant vesicles of ternary mixtures of phospholipids and cholesterol. *Biophysical journal*, 85(5), 3074-3083 (2003).

(92) Risselada, H. J. & Marrink, S. J. The molecular face of lipid rafts in model membranes. Proceedings of the National Academy of Sciences 105(45), 17367–17372 (2008).

(2) Both templates have been added simultaneously, with a 100-fold excess of the EGFR template. Was this the result of optimization? How is the kinetics of protein production? As EGFR is in far excess, a significant precipitation, at least in the early period of the reaction, due to limiting nanodiscs, should be expected. How is the oligomeric form of the inserted EGFR? Have multiple insertions into one nanodisc been observed?

We thank the reviewer for these insightful questions. Yes, the EGFR:ApoA1∆49 template ratio of 100:1 was empirically determined through optimization experiments now shown in the revised Supplementary Fig. 3. Cell-free reactions were performed across a range of EGFR:ApoA1∆49 template ratios (1:2 to 1:200) and sampled at different time points (2-19 hours). As shown in the gels, EGFR expression increased with higher template ratios and longer reaction times up to ~9 hours, while ApoA1 expression became clearly detectable only after 6 hours. Based on these results, we selected an EGFR:ApoA1∆49 ratio of 100:1 and 8-hour reaction time as the optimal condition, which yielded sufficient full-length EGFR incorporated into nanodiscs for ensemble and single-molecule experiments.

In cell-free systems, protein yield does not scale directly with DNA template concentration, as translation efficiency is limited by factors such as ribosome availability and co-translational membrane insertion [Hunt, Chem. Rev., 2024; Blackholly, Front. Mol. Biosci., 2022]. Consistent with this, we observed that ApoA1∆49 is produced at higher levels than EGFR despite the lower DNA input (Supplementary Fig. 2b). Providing an excess EGFR template prevents the reaction from becoming limited by scaffold availability and helps compensate for the fact that, as a large multi-domain receptor, EGFR expression can yield truncated as well as full-length products. This strategy ensures that sufficient full-length receptors are available for nanodisc incorporation. We will clarify this in the Methods section (see below).

We observed little to no visible precipitation under the reported cell-free conditions, likely due to the following reasons: (i) EGFR and ApoA1∆49 are co-expressed in the cell-free reaction, and ApoA1∆49 assembles into nanodiscs concurrently with receptor translation, providing an immediate membrane sink (ii) ApoA1∆49 is expressed at high levels, maintaining disc concentrations that keep the reaction in a soluble regime.

The sample contains donor-labeled EGFR (snap surface 594) together with acceptor-labeled lipids (cy5-labeled PE doped in the nanodisc). We assess the oligomerization state of EGFR in nanodiscs using single-molecule photobleaching of the donor channel. Snap surface 594 is a benzyl guanine derivative of Atto 594 that reacts with the SNAP tag with near-stoichiometry efficiency [Sun, Chembiochem, 2011]. Most molecules (~75%) exhibited a single photobleaching step, consistent with incorporation of a single EGFR per nanodisc [Srinivasan, Nat. Commun., 2022]. A minority of traces (~15%) showed two photobleaching steps and about ~10% of traces showed three or more photobleaching steps, consistent with occasional multiple insertions. For all smFRET analysis, we restricted the dataset to single-step photobleaching traces, ensuring measurements were performed on monomeric EGFR.

Methods, page 20, “Production of labeled, full-length EGFR nanodiscs: Briefly, the *E. coli* slyD lysate, in vitro protein synthesis *E. coli* reaction buffer, amino acids (-Methionine), Methionine, T7 Enzyme, protease inhibitor cocktail (Thermofisher Scientific), RNAse inhibitor (Roche) and DNA plasmids (20ug of EGFR and 0.2ug of ApoA1∆49) were mixed with different lipid mixtures. The DNA template ratio of EGFR:ApoA1∆49 = 100:1 was empirically chosen by testing different ratios on SDS-PAGE gels and selecting the condition that maximized full-length EGFR expression in DMPC lipids (Supplementary Fig. 3).”

(3) The IMAC purification does not discriminate between EGFR-filled and empty nanodiscs. Does the TEM study give any information about the composition of the particles (empty, EGFR monomers, or EGFR oligomers)? Normalizing the measured fluorescence, i.e., the total amount of solubilized receptor, with the total protein concentration of the samples could give some data on the stoichiometry of EGFR and nanodiscs.

Negative-stain TEM was performed to confirm nanodisc formation and morphology, but this method does not resolve whether a given disc contains EGFR. To directly assess receptor stoichiometry, we instead relied on single-molecule photobleaching of snap surface 594-labeled EGFR (see response to Point 2). These experiments showed that the majority of nanodiscs contain a single receptor, with a minority containing two receptors. For all smFRET analyses, we restricted data to single-step photobleaching traces, ensuring measurements were performed on monomeric EGFR.

We did not normalize EGFR fluorescence to total protein concentration because the bulk protein fraction after IMAC purification includes both receptor-loaded and empty nanodiscs. The latter contribute to ApoA1∆49 mass but do not contain receptors and including them would underestimate receptor occupancy. Importantly, the presence of empty nanodiscs does not affect our measurements as photobleaching and single-molecule FRET analyses selectively report only on receptor-containing nanodiscs. This clarification has been added to the Methods.

Methods, page 26, “Fluorescence Spectroscopy: Traces with a single photobleaching step for the donor and acceptor were considered for further analysis. Regions of constant intensity in the traces were identified by a change-point algorithm95. Donor traces were assigned as FRET levels until acceptor photobleaching. The presence of empty nanodiscs does not influence these measurements, as photobleaching and single-molecule FRET analyses selectively report on receptor-containing nanodiscs.”

(4) The authors generally assume a 100% functional folding of EGFR in all analyzed environments. While this could be the case, with some other membrane proteins, it was shown that only a fraction of the nanodisc solubilized particles are in functional conformation. Furthermore, the percentage of solubilized and folded membrane protein may change with the membrane composition of the supplied nanodiscs, while non-charged lipids mostly gave rather poor sample quality. The authors normalize the ATP binding to the total amount of detectable EGFR, and variations are interpreted as suppression of activity. Would the presence of unfolded EGFR fractions in some samples with no access to ATP binding be an alternative interpretation?

We agree that not all nanodisc-embedded EGFR molecules may be fully functional and that the fraction of folded protein could vary with lipid composition. In our ATP-binding assay, EGFR detection relies on the C-terminal SNAP-tag fused to an intrinsically disordered region. Successful labeling requires that this segment be translated, accessible, and folded sufficiently to accommodate the SNAP reaction, which imposes an additional requirement compared to the rigid, structured kinase domain where ATP binds. Misfolded or truncated EGFR molecules would therefore likely fail to label at the C-terminus. These factors strongly imply that our assay predominantly reports on receptor molecules that are intact and well folded.

Additionally, our molecular dynamics simulations at 0% and 30% POPS support the experimental ATP-binding measurements (Fig. 2c, d). This consistency between both the experimental and simulated evidence, including at 0% POPS where reduced receptor folding might be expected, suggests that the observed lipid-dependent changes are more likely due to modulation of the functional receptor rather than receptor misfolding. We have clarified these points by adding the following

Results, page 7, “Role of anionic lipids in EGFR kinase activity: In the presence of EGF, increasing the anionic lipid content decreased the number of contacts from 71.8 ± 1.8 to 67.8 ± 2.4, indicating increased accessibility, again in line with the experimental findings. Because detection of EGFR relies on labeling at the C-terminus and ATP binding requires an intact kinase domain, the ATPbinding assay is for receptors that are properly folded and competent for nucleotide binding. The consistency between experimental results and MD simulations suggests that the observed lipiddependent changes are more likely due to modulation of functional EGFR than to artifacts from misfolding.”

**Reviewer #1 (Recommendations for the authors):**
The experimental program presented here is excellent, and the results are highly interesting. My enthusiasm is dampened by the presentation in places which is confusing, especially Figure 3, which contains so many of the results. I also have some reservations about the bimodal interpretation of the lifetime data in Figure 3.

We thank the reviewer for their positive assessment of our experimental approach and results. In the revised version, we have improved figure organization and readability by adding explicit labels for lipid composition and EGF presence/absence in all lifetime distributions, moving key supplementary tables into main text, and reorganizing the supplementary figures as Extended Data Figures following *eLife’s* format. Figures and tables now appear in the order in which they are referenced in the text to further improve readability.

Regarding the bimodal interpretation of the lifetime distribution, we have performed global fits of the data with both two- and three-Gaussian models and evaluated them using the Bayesian Information Criterion (BIC) and Ashman’s D analysis, which supported the bimodal interpretation. Details of this analysis are provided in our response to comment (8) below and included in the manuscript.

Specific comments below:(1) Abstract -"Identifying and investigating this contribution have been challenging owing to the complex composition of the plasma membrane" should be "has".

We have corrected this error in the revised manuscript.

(2) Results - p4 - some explanation of what POPC/POPS are would be helpful.

We have added the text below discussing POPC and POPS.

Results, page 4, “POPC is a zwitterionic phospholipid forming neutral membranes, whereas POPS carries a net negative charge and provides anionic character to the bilayer[56]. Both PC and PS lipids are common constituents of mammalian plasma membranes, with PC enriched in the outer leaflet and PS in the inner leaflet[22].”

(22) Lorent, J. H., Levental, K. R., Ganesan, L., Rivera-Longsworth, G., Sezgin, E., Doktorova, M., Lyman, E. & Levental, I. Plasma membranes are asymmetric in lipid unsaturation, packing and protein shape. Nature Chemical Biology 16, 644–652 (2020).

(56) Her, C., Filoti, D. I., McLean, M. A., Sligar, S. G., Ross, J. A., Steele, H. & Laue, T. M. The charge properties of phospholipid nanodiscs. Biophysical journal 111(5), 989–998 (2016).

(3) Figure 2b - it would be easier to compare if these were plotted on top of each other. Are we at saturating ATP binding concentration or below it? Also, please put a key to say purple - absent and orange +EGF on the figure. I am also confused as to why, with no EGF, ATP binding is high with 0% POPS, but low when EGF is present, but that then reverses with physiological lipid content.

While we agree that a direct comparison would be easier, the ATP-binding experiments for the ± EGF conditions were actually performed independently on separate SDS-PAGE gels, which unfortunately precludes such a comparison. We have added a color key to clarify the -EGF and +EGF datasets.

The experiments were carried out at 1 µM of the fluorescently labeled ATP analogue (atto647Nγ ATP). Reported kinetic measurements for the isolated EGFR kinase domain indicate an K_m_ of 5.2 µM suggesting that our experimental concentration is below, but close to the saturating range ensuring sensitivity to changes in accessibility of the binding site rather than saturating all available receptors.

We have revised the manuscript to clarify these details by including the following text:

Results, page 6, “To investigate how the membrane composition impacts accessibility, we measured ATP binding levels for EGFR in membranes with different anionic lipid content. 1 µM of fluorescently-labeled ATP analogue, atto647N-γ ATP, which binds irreversibly to the active site, was added to samples of EGFR nanodiscs with 0%, 15%, 30% or 60% anionic lipid content in the absence or presence of EGF.”

Methods, page 24, “ATP binding experiments: Full-length EGFR in different lipid environments was prepared using cell-free expression as described above. 1μM of snap surface 488 (New England Biolabs) and atto647N labeled gamma ATP (Jena Bioscience) was added after cell-free expression and incubated at 30 °C , 300 rpm for 60 minutes. 1μM of atto647N-γ ATP was used, corresponding to a concentration near the reported Km of 5.2 µM for ATP binding to the isolated EGFR kinase domain[93], ensuring sensitivity to lipid-dependent changes in ATP accessibility.”

(ii) Nucleotide binding is suppressed under basal conditions, likely to ensure that the catalytic activity is promoted only upon EGF stimulation.

The molecular dynamics simulations at 0% and 30% POPS further support this interpretation, showing that anionic lipids modulate the accessibility of the ATP-binding site in a manner consistent with experimental trends (Fig. 2c and 2d).

We have clarified these points in the main text with the following additions:

Results, page 6, “In the presence of EGF, ATP binding overall increased with anionic lipid content with the highest levels observed in 60% POPS bilayers. In the neutral bilayer, ligand seemed to suppress ATP binding, indicating anionic lipids are required for the regulated activation of EGFR.”

Results, page 7, “In the absence of EGF, increasing the anionic lipid content from 0\% POPS to 30% POPS increased the number of ATP-lipid contacts 58.6±0.7 to 74.4±1.2, indicating reduced accessibility, consistent with the experimental results and suggesting anionic lipids are required for ligand-induced EGFR activity.”

(93) Yun, C. H., Mengwasser, K. E., Toms, A. V., Woo, M. S., Greulich, H., Wong, K. K., Meyerson,M. & Eck, M.J. The T790M mutation in EGFR kinase causes drug resistance by increasing the affinity for ATP. PNAS, 105(6), 2070–2075 (2008).

(4) Figure 2d - how was the 16A distance arrived at?

We thank the reviewer for pointing this out. The 16 Å cutoff was chosen based on the physical dimensions of the ATP analogue used in the experiments. Specifically, the largest radius of the atto647N-γ ATP molecule is ~16.9 Å, which defines the maximum distance at which lipid atoms could sterically obstruct access of ATP to the binding pocket. Accordingly, in the simulations, contacts were defined as pairs of coarse-grained atoms between lipid molecules and the residues forming the ATP-binding site (residues 694-703, 719, 766-769, 772-773, 817, 820, and 831) separated by less than 16 Å.

We have rewritten the rationale for selecting the 16 Å cutoff in the Methods section to improve clarity.

Methods, page 28, “Coarse-grained, Explicit-solvent Simulations with the MARTINI Force Field: We analyzed our simulations using WHAM[108,109] to reweight the umbrella biases and compute the average values of various metrics introduced in this manuscript. Specifically, we calculated the distance between Residue 721 and Residue 1186 (EGFR C-terminus) of the protein. To quantify the accessibility of the ATP-binding site, we calculated the number of contacts between lipid molecules and the residues forming the ATP-binding pocket (residues 694-703, 719, 766-769, 772-773, 817, 820, and 831)[110]. Close contact between the bilayer and these residues would sterically hinder ATP binding; thus, the contact number serves as a proxy for ATP-site accessibility. The cutoff distance for defining a contact was set to 16 Å, corresponding to the largest molecular radius of the fluorescent ATP analogue (atto647N-γ ATP, 16.96 Å111). Accordingly, we defined a contact as a pair of coarse-grained atoms, one from the lipid membrane and one from the ATP binding site, within a mutual distance of less than 16 Å.”

(5) Figure 2e-h - I think a bar chart/violin plot/jitter plot would make it easier to compare the peak values. The statistics in the table should just be quoted in the text as value +/- error from the 95% confidence interval. The way it is written currently is confusing, as it implies that there is no conformational change with the addition of EGF in neutral lipids, but there is ~0.4nm one from the table. I don't understand what you mean by "The larger conformational response of these important domains suggests that the intracellular conformation may play a role in downstream signaling steps, such as binding of adaptor proteins"?

We thank the reviewer for these suggestions. For the smFRET lifetime distributions (Figure 2j, k; previously Figure 2e, f), we have now included jitter plots of the donor lifetimes in the Supplementary Figure 11 to facilitate direct visual comparison of the median and distribution widths for each lipid composition and ±EGF conditions. The distance distributions for the ATP to C-terminus in Figure 2e, f (previously Figure 2g, h) were obtained from umbrella-sampling simulations that calculate free-energy profiles rather than raw, unbiased distance values. Because the sampling is guided by biasing potentials, individual distance values cannot be used to construct violin or jitter plots. We therefore present the simulation data only as probability density distributions, which best reflect the equilibrium distributions derived from them.

We have also revised the text to report the median ± 95% confidence interval, improving clarity and consistency with the statistical table.

Results, page 9: “In the neutral bilayer (0% POPS), the distributions in the absence of EGF peaks at 8.1 nm (95% CI: 8.0–8.2 nm) and in the presence of EGF peaks at 8.6 nm (95% CI: 8.5–8.7 nm) (Table 1, Supplementary Table 1). In the physiological regime of 30% POPS nanodiscs, the peak of the donor lifetime distribution shifts from 9.1 nm (95% CI: 8.9–9.2 nm) in the absence of EGF to 11.6 nm (95% CI: 11.1–12.6 nm) in the presence of EGF (Table 1, Supplementary Table 1), which is a larger EGF-induced conformational response than in neutral lipids.”

Finally, we have rephrased the sentence in question for clarity. The revised text now reads:

Results, page 9: “The larger conformational response observed in the presence of anionic lipids suggests that these lipids enhance the responsiveness of the intracellular domains to EGF, potentially ensuring interactions between C-terminal sites and adaptor proteins during downstream signaling.”

(6) "r, highlighting that the charged lipids can enhance the conformational response even for protein regions far away from the plasma membrane" - is it not that the neutral membrane is just very weird and not physiological that EGFR and other proteins don't function properly?

We agree with the reviewer that completely neutral (0% POPS) membranes are not physiological and likely do not support the native organization or activity of EGFR. We have revised the text to clarify that the 30% POPS condition represents a more native-like lipid environment that restores or stabilizes the expected conformational response, rather than "enhancing" it. The revised sentence now reads:

Results, page 10: “Both experimental and computational results show a larger EGF-induced conformational change in the partially anionic bilayer, consistent with the notion that a partially anionic lipid bilayer provides a more native environment that supports proper receptor activation, compared to the non-physiological neutral membrane.”

(7) "snap surface 594 on the C-terminal tail as the donor and the fluorescently-labeled lipid (Cy5) as the acceptor (Supplementary Fig. 2, 11)." Why not refer to Figure 3a here to make it easier to read?

We have added the reference to Figure 3a, and we thank the Reviewer for the suggestion.

(8) Figure 3 - the bimodality in many of these plots is dubious. It's very clear in some, i.e. 0% POPS +EGF, but not others. Can anything be done to justify bimodality better?

We agree that statistical justification is important for interpreting lifetime distributions. To address this, we performed global fits of the data with both two- and three-Gaussian models and evaluated them using the Bayesian Information Criterion (BIC), which balances the model fit with a penalty for additional parameters. The three-Gaussian model gave a substantially lower BIC, indicating statistical preference for the more complex model. However, we also assessed the separability of the Gaussian components using Ashman’s D, which quantifies whether peaks are distinct. This analysis showed that two of the Gaussians are not separable, implying they represent one broad distribution rather than two discrete states. Therefore, when all the distributions are fit globally, the data are best described as two Gaussians, one centered at ~1.3 ns and the other at ~2.7 ns, with cholesterol-dependent shifts reflecting changes in the distribution of this population rather than the emergence of a separate state. We better justified our choice of model by incorporating the results of the global two- vs three-Gaussian fits with BIC and Ashman’s D analysis in the revised manuscript.

Methods, page 27: “Model Selection and Statistical Analysis

Global fitting of lifetime distributions was performed across all experimental conditions using maximum likelihood estimation. Both two-Gaussian and three-Gaussian distribution models were evaluated as described previously.62 Model performance was compared using the Bayesian Information Criterion (BIC),[101] which balances model likelihood and complexity according to

BIC = -2 ln L + k ln n

where L is the likelihood, k is the number of free parameters, and n is the number of singlemolecule photon bunches across all experimental conditions. A lower BIC value indicates a statistically better model[101]. The separation between Gaussian components was subsequently assessed using the Ashman’s D where a score above 2 indicates good separation[102]. For two Gaussian components with means µ1, µ2 and standard deviations σ1, σ2,\begin{document}$$\displaystyle D_{i j}=\frac{\left|\mu_{\mathrm{i}}-\mu_{\mathrm{j}}\right|}{\sqrt{\frac{1}{2}\left(\sigma_{i}^{2}+\sigma_{j}^{2}\right)}}$$\end{document}

where Dij represents the distance metric between Gaussian components i and j. All fitted parameters, likelihood values, BIC scores, and Ashman’s D values are summarized in Supplementary Table 5.”

(101) Schwarz, G. Estimating the dimension of a model. The Annals of Statistics, 461–464 (1978).

(102) Ashman, K. M., Bird, C. M. & Zepf, S. E. Detecting bimodality in astronomical datasets. The Astronomical Journal 108(6), 2348–2361 (1994).

(9) Figure 3c - can you better label the POPS/POPC on here?

We thank the reviewer for this suggestion. In the revised manuscript, Figure 3b (previously Figure 3c) has been updated to label the lipid composition corresponding to each smFRET distribution to make the comparison across conditions easier to follow.

(10) Figure 3g - it looks like cholesterol causes a shift in both the peaks, such that the previous open and closed states are not the same, but that there are 2 new states. This is key as the authors state: "Remarkably, high anionic lipids and cholesterol content produce the same EGFR conformations but with opposite effects on signaling-suppression or enhancement." But this is only true if there really are the same conformational states for all lipid/cholesterol conditions. Again, the bimodal models used for all conditions need to be justified.

We appreciate the reviewer’s insightful comment. We agree that the interpretation of the lifetime distributions depends on whether cholesterol and anionic lipids modulate existing conformational states or create new ones. To test this, we performed global fits of all distributions using the two- and three-Gaussian models and compared them using the Bayesian Information Criterion (BIC) and Ashman’s D, the results of which are described in detail in response to (8) above.

Both fitting models, two- and three-Gaussian, identified the same short lifetime component (µ = 1.3 ns), suggesting this reflects a well separated conformation. While the three-Gaussian model gave a lower BIC, Ashman’s D analysis indicated that the two of the three components (µ = 2.6 ns and 3.4 ns) are not statistically separable, suggesting they represent a single broad conformational population rather than distinct states. If instead these two components reflected distinct states present under different conditions, Ashman’s D analysis would have found the opposite result. This supports our interpretation that high cholesterol and high anionic lipid content produce similar conformation ensembles with opposite effects on signaling output.

Finally, we acknowledge that additional conformations may exist, but based on this analysis a bimodal model describes the populations captured in our data and so we limit ourselves to this simplest framework. We have clarified this rationale in the revised manuscript and added the results of the BIC and Ashman’s D analysis to support this interpretation.

(11) Why are we jumping about between figures in the text? Figure 1d is mentioned after Figure 2. Also, DMPC is shown in the figures way before it is described in the text. It is very confusing. Figure 3 is so compact. I think it should be spread out and only shown in the order presented in the text. Different parts of the figure are referred to seemingly at random in the text. Why is DMPC first in the figure, when it is referred to last in the text?

Following the Reviewer’s comment, we have revised the figure order and layout to improve readability and ensure consistency with the text. The previous Figures 1d-f which introduce the single-molecule fluorescence setup are now Figure 2g-i, positioned immediately before the first single-molecule FRET experiments (Fig 2j, k). The DMPC distribution in Figure 3 has been moved to the Supplementary Information (Supplementary Fig. 17), where it is shown alongside POPC, as these datasets are compared in the section “Mechanism of cholesterol inhibition of EGFR transmembrane conformational response”. The smFRET distributions in Figure 3 are now presented in the same sequence as they are discussed in the text, and the figure has been spread out for better clarity.

(12) Throughout, I find the presentation of numerical results, their associated error, and whether they are statistically significantly different from each other confusing. A lot of this is in supplementary tables, but I think these need to go in the main text.

To improve clarity and ensure that key quantitative results are easily accessible, we have moved the relevant supplementary tables to the main text. Specifically, the following tables have been incorporated into the main manuscript:

(i) Median distance between the ATP binding site and the EGFR C-terminus, or between membrane and EGFR C-terminus from smFRET measurements (previously supplementary table 1 is now main table 1)

(ii) Median distance between the membrane and the EGFR C-terminus in different anionic lipid environments (previously supplementary table 4 is now main table 2)

(iii) Median distance between the membrane and the EGFR C-terminus in different cholesterol environments (previously supplementary table 8 and 12 is now combined to be main table 3)

(13) Supplementary figures - in general, there is a need to consider how to combine or simplify these for eLife, as they will have to become extended data figures.

We thank the reviewer for this helpful suggestion. In the revised manuscript, we have reorganized the supplementary figures into extended data figures in accordance with eLife’s format. Specifically:

- Supplementary Figs. 1–7 are now grouped as Extended Data Figures for Figure 1 in the main text. They are now Figure 1 - figure supplements 1–7.

- Supplementary Fig. 8–11 is now Extended Data Figure associated with Figure 2. It is now Figure 2 - figure supplements 1–4.

- Supplementary Figs. 12–17 are now grouped as Extended Data Figures for Figure 3. They are now Figure 3 - figure supplements 1–6.

(14) Supplementary Figure 2 - label what the two bands are in the EGFR and pEGFR sets at the bottom of panel c.

We thank the reviewer for this comment. The two bands shown in the EGFR and pEGFR blots in Supplementary Fig. 2d (previously Supplementary Fig. 2c) corresponds to replicate samples under identical conditions. We have now clarified this in the figure legend and labeled the lanes as “Rep 1” and “Rep 2” in the revised figure and modified the figure legend.

Supplementary Figure 2, page 31: “(d) Western blots were performed on labelled EGFR in nanodiscs. Anti-EGFR Western blots (left) and anti-phosphotyrosine Western blots (right) tested the presence of EGFR and its ability to undergo tyrosine phosphorylation, respectively, consistent with previous experiments on similar preparations[18, 54, 55]. The two lanes in each blot correspond to replicate samples under identical conditions.”

(15) Supplementary Figures 3+4 - a bar chart/boxplot or similar would be easier for comparison here.

In the revised version, we have replaced the histograms with jitter plots showing the nanodisc size distributions for each condition in supplementary figures 4 and 5 (previously supplementary figures 3 and 4). The plots display individual measurements with a horizontal line indicating the mean size (mean ± standard deviation values provided in the caption).

(16) Supplementary Figures 10, 12, 13, 15, 16 - I would jitter these.

We have incorporated jitter plots for the relevant datasets in Supplementary Figures 11, 13, 15, 16 and 17 (previously supplementary figures 10, 12 13, 15 and 16) to provide a clearer visualization of the data distributions and median values.

**Reviewer #2 (Recommendations for the authors):**
(1) Reactions were performed in 250 µL volumes. What is the average yield of solubilized EGFR in those reactions? Are there differences in the EGFR solubilization with the various lipid mixtures?

The amount of solubilized EGFR produced in each 250 µL cell-free reaction was below the reliable detection limit for quantitative absorbance assays. At these protein levels, little to no EGFR precipitation was observed for all lipid compositions. Although exact yields could not be determined, fluorescence-based detection confirmed the presence of functional, nanodiscincorporated EGFR suitable for smFRET and ensemble fluorescence experiments. We observed variability in total yield between independent reactions within the same lipid composition, which is common for cell-free systems, but no consistent trend attributable to lipid composition.

(2) Figure S2: It would be better to have a larger overview of the particles on a grid to get a better impression of sample homogeneity.

TEM images showing a larger field of view have been added for each lipid composition in Supplementary Figures 4 and 5.

(3) Figure 2b: It appears that there is some variation in the stoichiometry of ApoA1 and EGFR within the samples. Have equal amounts of each sample been analyzed? Are there, in addition, some precipitates of EGFR? It would further be good to have a negative control without expression to get more information about the additional bands in Figure S2b. As they do not appear in the fluorescent gel, it is unlikely that they represent premature terminations of EGFR.

The fluorescence intensity from the bound ATP analogue (Atto 647N-ATP) and from the snap surface 488 label, which binds stoichiometrically to the SNAP tag at the EGFR C-terminus, was measured for each sample. The relative amount of ATP binding was quantified for each sample by normalizing to the EGFR content (Figure 2b). This normalization accounts for the different amounts of EGFR produced in each condition.

We did not observe any visible precipitation under the reported cell-free conditions, likely due to the following reasons:

(i) EGFR and ApoA1 are co-expressed in the cell-free reaction, and ApoA1 assembles into nanodiscs concurrently with receptor translation, providing an immediate membrane sink

(ii) ApoA1 is expressed at high levels, maintaining disc concentrations that keep the reaction in a soluble regime.

A control cell-free reaction containing only ApoA1∆49 (1 µg) and no EGFR template, analyzed after affinity purification, showed a single prominent band at ~ 25 kDa (gel image below), corresponding to ApoA1, along with faint background bands typical of Ni-NTA purification from cell-lysates. These weak, non-specific bands likely arise from co-purification of endogenous *E. coli* proteins.

The ApoA1∆49-only control gel has now been included as part of the supplementary figure 2.

(4) Figure S2c: It would be better to show the whole lanes to document the specificity of the antibodies. Anti-Phosphor antibodies are frequently of poor selectivity. In that case, a negative control with corresponding tyrosine mutations would be helpful.

We have updated Figure S2d (previously Figure S2c) to include the full gel lanes to better illustrate the specificity of both the total EGFR and phospho-EGFR (Y1068) antibodies. The results show a single clear band at the expected molecular weight for EGFR, conforming antibody specificity.

(5) The Results section already contains quite some discussion. I would thus recommend combining both sections.

We thank the reviewer for the suggestion. We have now created a results and discussion section to better reflect the content of these paragraphs, with the previous discussion section now a subsection focused on implications of these results.